# Correcting for sparsity and interdependence in glycomics by accounting for glycan biosynthesis

Bokan Bao[1,2,3,6], Benjamin P. Kellman [1,2,3,6], Austin W. T. Chiang [1,4], Yujie Zhang [1], James T. Sorrentino [1,2,3], Austin K. York[1], Mahmoud A. Mohammad [5], Morey W. Haymond[5], Lars Bode[1] & Nathan E. Lewis [1,3,4 ✉]

Glycans are fundamental cellular building blocks, involved in many organismal functions. Advances in glycomics are elucidating the essential roles of glycans. Still, it remains challenging to properly analyze large glycomics datasets, since the abundance of each glycan is dependent on many other glycans that share many intermediate biosynthetic steps. Furthermore, the overlap of measured glycans can be low across samples. We address these challenges with GlyCompare, a glycomic data analysis approach that accounts for shared biosynthetic steps for all measured glycans to correct for sparsity and non-independence in glycomics, which enables direct comparison of different glycoprofiles and increases statistical power. Using GlyCompare, we study diverse N-glycan profiles from glycoengineered erythropoietin. We obtain biologically meaningful clustering of mutant cell glycoprofiles and identify knockout-specific effects of fucosyltransferase mutants on tetra-antennary structures. We further analyze human milk oligosaccharide profiles and find mother's fucosyltransferase-dependent secretor-status indirectly impact the sialylation. Finally, we apply our method on mucin-type O-glycans, gangliosides, and site-specific compositional glycosylation data to reveal tissues and disease-specific glycan presentations. Our substructure-oriented approach will enable researchers to take full advantage of the growing power and size of glycomics data.

[1] Department of Pediatrics, University of California, San Diego, La Jolla, CA, USA. [2] Bioinformatics and Systems Biology Graduate Program, University of California, San Diego, La Jolla, CA, USA. [3] Department of Bioengineering, University of California, San Diego, La Jolla, CA, USA. [4] The Novo Nordisk Foundation Center for Biosustainability at the University of California, San Diego, La Jolla, CA, USA. [5] Department of Pediatrics, Children's Nutrition Research Center, US Department of Agriculture/Agricultural Research Service, Baylor College of Medicine, Houston, TX, USA. [6]These authors contributed equally: Bokan Bao, Benjamin P. Kellman. ✉email: nlewisres@ucsd.edu

Glycosylation is a complex post-translational modification and it decorates one-fifth to one-half of eukaryotic proteins[1,2]. Glycans account for 12–25% of dry cell mass and have essential functional and pathological roles[3,4]. Despite their importance, glycans have complex structures that are difficult to study. The complex structures of glycans arise from a context-sensitive biosynthetic network involving dozens of enzymes. A simple change of a single intermediate glycan or glycosyltransferase will have cascading impacts on the glycans secreted[5,6]. Unfortunately, current data analysis approaches for glycoprofiling and glycomic data lack the critical systems perspective to decode the interdependence of glycans easily[7–12]. It is important to understand the network behind the glycoprofiles to understand the behavior of the process better.

New tools aiding in the acquisition and aggregation of glycoprofiles are emerging, making large-scale comparisons of glycoprofiles possible. Advances in mass spectrometry now enable the rapid generation of many glycoprofiles with detailed glycan composition and structure predictions[7,8,10–18], exposing the complex and heterogeneous glycosylation patterns on lipids and proteins[8,9,11,19–22]. Large glycoprofile datasets and supporting databases are also emerging, including GlyTouCan[23], UniCarb-DB[24], GlyGen[25], and UniCarbKB[26].

These technologies and databases enable efforts to associate glycans with disease and other phenotypes. However, the rapid and accurate comparison of glycoprofiles can be challenging with the size, sparsity and heterogeneity of such datasets[8,9,11,20–22,27]. A glycoprofile provides glycan structure and abundance information, and each glycan is usually treated as an independent entity. Furthermore, in any one glycoprofile, only a small percentage of all possible glycans may be detected[8,9,11,20,27]. Thus, if there is a significant perturbation to glycosylation in a dataset, only a few glycans, if any, may overlap between samples. However, these non-overlapping glycans may only differ in their synthesis by as few as one enzymatic step. Currently, deliberate manual coding is required to make glycoprofiles comparable[7–12,20,27]. These properties of glycomics data may not be problematic in the studies of individual glycans and their downstream effects on other biological processes. However, sparsity and non-independence can be a problem in determining the sources of differential glycan abundance when leveraging large datasets[8,9,27,28]. Since many methods assume data independence (e.g., $t$-tests, ANOVA, etc.), their application to glycomics can lead to decreased statistical power or erroneous results.

Previous studies have used glycan motifs to explore similarities across glycans. Scientists have used substructure-oriented analysis to describe glycan diversity in databases (e.g., glycan fingerprinting)[29,30], align glycan structures[31], identify glycan epitopes in glycoprofiles[32] and lectin profiles[1], deconstruct LC-MS data to quantify glycan abundance[33], and compare glycans in glycoprofiles[34]. These tools leverage both glycan abundance and structure. However, no tools make explicit use of the biosynthetic context encoded in glycan structures. Thus, a generalized substructure approach could facilitate the study of large numbers of glycoprofiles by connecting them to the shared mechanisms involved in making each glycan.

In this work we present GlyCompare, a method enabling the rapid and scalable analysis and comparison of multiple glycoprofiles, while accounting for the biosynthetic similarities of each glycan. We propose glycan substructures, or intermediates, as interpretable functional units for glycoprofile comparisons; each substructure can capture one step in the complex process of glycan biosynthesis, which accounts for the shared dependencies across glycans. This approach addresses current challenges in sparsity and hidden interdependence across glycomic samples and will facilitate discovering mechanisms underlying the changes among glycoprofiles. We demonstrate the functionality and performance of this approach with a variety of glycomic analyses, including recombinant erythropoietin (EPO) N-glycosylation, human milk oligosaccharides (HMOs), mucin-type O-glycans, gangliosides, and site-specific compositional data. Specifically, we analyzed 16 MALDI-TOF glycoprofiles of EPO, where each EPO glycoprofile was produced in a different glycoengineered CHO cell line[21,27]. We also analyze 48 HPLC glycoprofiles of HMO from six mothers[35]. By analyzing these glycoprofiles with GlyCompare, we quantify the abundance of important substructures, cluster the glycoprofiles of mutant cell lines, connect genotypes to unexpected changes in glycoprofiles, and associate a phenotype of interest with substructure abundance and flux. We further demonstrate that such analyses gain statistical power. Finally, we expand our studies to include a tumor-normal comparison of mucin-type O-glycans, human retinal glycolipids, and site-specific N-glycan compositional data from the mouse brain. The analyses of the various N- and O-type glycan datasets demonstrate that our framework presents a convenient and automated approach to elucidate insights into complex patterns in glycobiology.

## Results

**Different glycoprofiles from small genetic changes can be compared with GlyCompare.** Due to the sparsity and interdependence of glycans in each glycoprofile, comparing different glycoprofiles can be challenging[9,12]. We demonstrated the core idea with three diverse erythropoietin (EPO) profiles made by three glycoengineered CHO cell lines[21,27]. EPO produced in the wild type (WT) and two double glycosyltransferase knockout (Mgat4a/4b and St3gal3/6) CHO cell lines have very different glycoprofiles that do not share many detected glycans (Fig. 1a). Efforts to identify primary and off-target effects of genetic modifications have limited success when relying only on overlapping glycans or on the presence/absence of a set of glycoforms. This glycan-level analysis can drastically limit analytic power due to the sparsity of comparable consensus glycans (Fig. 1a). The problem is that even glycans differing in only one single monosaccharide will be treated as two completely different glycans under conventional glycoprofile analysis methods[8]. Ultimately, the glycan abundance cannot be compared directly. This limited overlap between samples compounds when analyzing large glycomics datasets. These challenges prompted us to develop GlyCompare, a substructure-based approach to glycan analysis. Glycoprofiles are decomposed into a substructure network that encodes the shared biosynthetic pathways as well as the interdependence among glycans. Then, the substructure abundances are aggregated across glycans to account for activities at each enzymatic step (Fig. 1b). In essence, this shifts the focus of glycoprofile analysis from examining the increase/decrease of independent glycans to examining the increase/decrease of a series of glycan substructures (Fig. 1c). Substructure abundance provides interpretable biosynthetic information and allows us to mitigate major statistical challenges of working with glycan-based glycoprofiles.

**GlyCompare decomposes glycoprofiles to facilitate comparison.** Glycoprofiles can be decomposed into abundances of intermediate substructures. The resulting substructure profile has richer information than whole glycan profiles and enables more precise comparison across conditions. Since glycan biosynthesis involves long, redundant pathways, the pathways can be collapsed to obtain a subset of substructures while preserving the information of all glycans in the dataset. We call this minimal set of

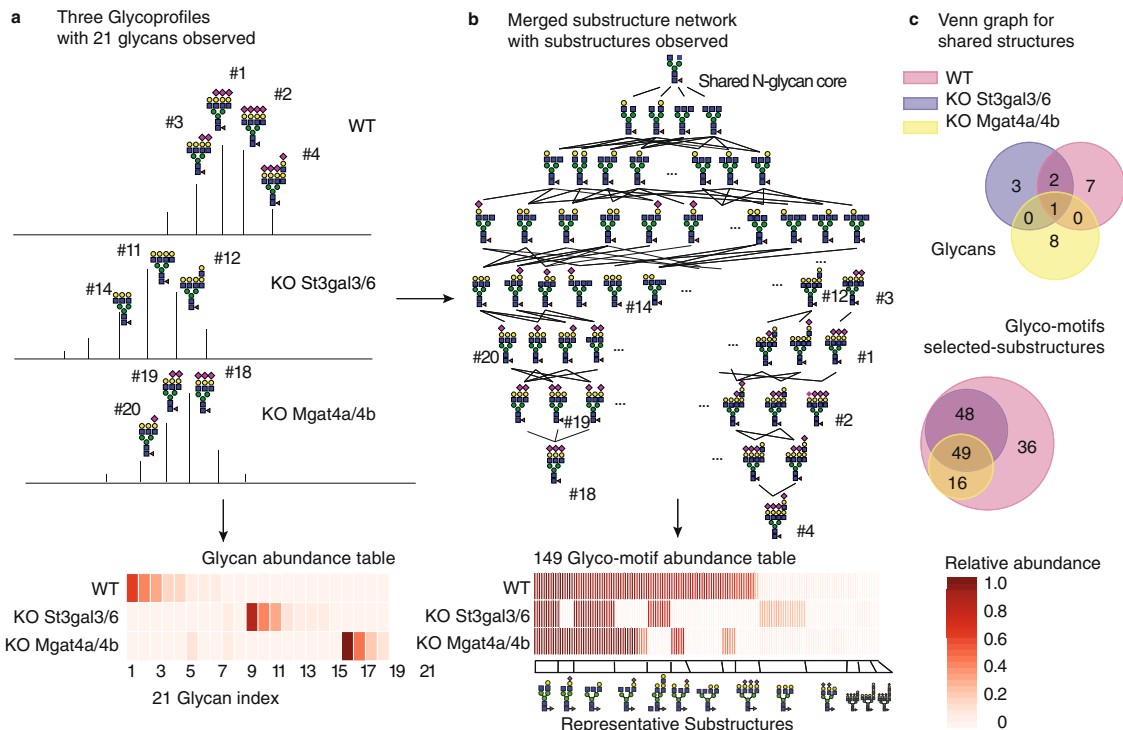

**Fig. 1 The GlyCompare platform improves glycomics data analysis and interpretation by using glycan biosynthetic network data to account for glycan interdependence. a** Three example glycoprofiles (WT, Mgat4a/4b knockout, and St3gal4/6 knockout profiles), with annotated glycans and measured relative glycan abundances, show low overlap despite differing in only a few gene knockouts. **b** The low overlap can be rescued by propagating glycan substructures through the glycan biosynthetic network. Then, the glycoprofile is transformed into glyco-motif vectors. The representative substructure is generated to represent core glycan substructures of glycoprofiles (see "Methods"). **c** Venn diagrams show the imperfect overlap of glycans across samples (upper), which is rescued when using GlyCompare to analyze glyco-motif substructures (bottom). Source data are provided as a Source data file.

substructures, glyco-motifs. The GlyCompare workflow consists of several steps wherein glycoprofiles are annotated and decomposed, glyco-motifs are prioritized, and each glyco-motif is quantified for subsequent comparisons (see the "Methods" section). The specific workflow is described as follows.

First, to characterize one glycoprofile with substructures, all substructures in one glycoprofile are identified and occurrence per glycan is quantified (Fig. 2a, b). Within a glycoprofile, a substructure's abundance is calculated by summing the abundance of all glycans containing the substructure. This transformation results in a substructure profile, which stores abundances for all glycan substructures (Fig. 2b) in the given glycoprofile. The summation over similar structures asserts that they follow the same synthetic paths, which is appropriate for glycosylation wherein synthesis is hierarchical and acyclic[6]. Therefore, a substructure abundance is not simply a sum over similar structures but mirrors the activity of the enzymes through biosynthetic pathways. Second, to identify the most informative substructures (i.e., glyco-motifs), substructures are prioritized using the substructure network. The substructure network is built by connecting all substructures with biosynthetic steps (Figs. 2d and 3c). The network starts from a core structure. An additional network level represents one biosynthetic step, adding one of the monosaccharides to the previous level. The edges in the network represent enzymatic additions of each monosaccharide, which can be annotated with known reactions (Supplementary Fig. 1). Redundant substructures are identified when parent–child substructure abundances are the same (Fig. 2d). Substructure network reduction proceeds by collapsing links with redundant substructures (connected with a solid arrow in Fig. 2d) and only retaining the child substructure. The remaining substructures are called glyco-motifs (selected-substructures); they describe the

variance entirely at the substructure level. The abundances of all glyco-motifs are then represented as a glyco-motif profile, the minimal subset of meaningful substructure abundances representing glycoprofiles (Fig. 2e).

For larger datasets, it is useful to summarize the structure difference and abundance changes by clustering glyco-motifs (Supplementary Fig. 2). After clustering glyco-motifs, the common structural features of a cluster are calculated using the average weight of each monosaccharide (Fig. 2f, see "Methods"). Monosaccharides with a weight larger than 51% are preserved, which illustrates the predominant structure in the cluster. This allows one to quickly evaluate the distinguishing structure features that vary across samples in any given dataset.

The workflow described here will connect all glycoprofiles in a dataset through their shared intermediate substructures, thus allowing robust analysis of the differences across glycomics samples and the evaluation of the associated genetic bases.

**GlyCompare accurately clusters glycoengineered EPO samples.** We first apply GlyCompare on the dataset consisting of 16 glycoprofiles coming from a panel of different erythropoietin (EPO) glycoforms, each produced in different glycoengineered CHO cell lines. Clustering glycoprofiles did not adequately recapitulate the severity of glycosylation disruption, wherein many neighboring samples were not the most genetically similar mutants (Fig. 3a and Supplementary Fig. 3). This inconsistency and poor clustering stem from the inherent sparseness of glycoprofiles, i.e., each glycoprofile only has a few observed glycans (Fig. 3d), and most glycans appear only in a few glycoprofiles (Figs. 3e, f). Thus, the matrix of glycan abundances is sparse and incompatible with the glycan synthesis assumption. Since glycan composition is not

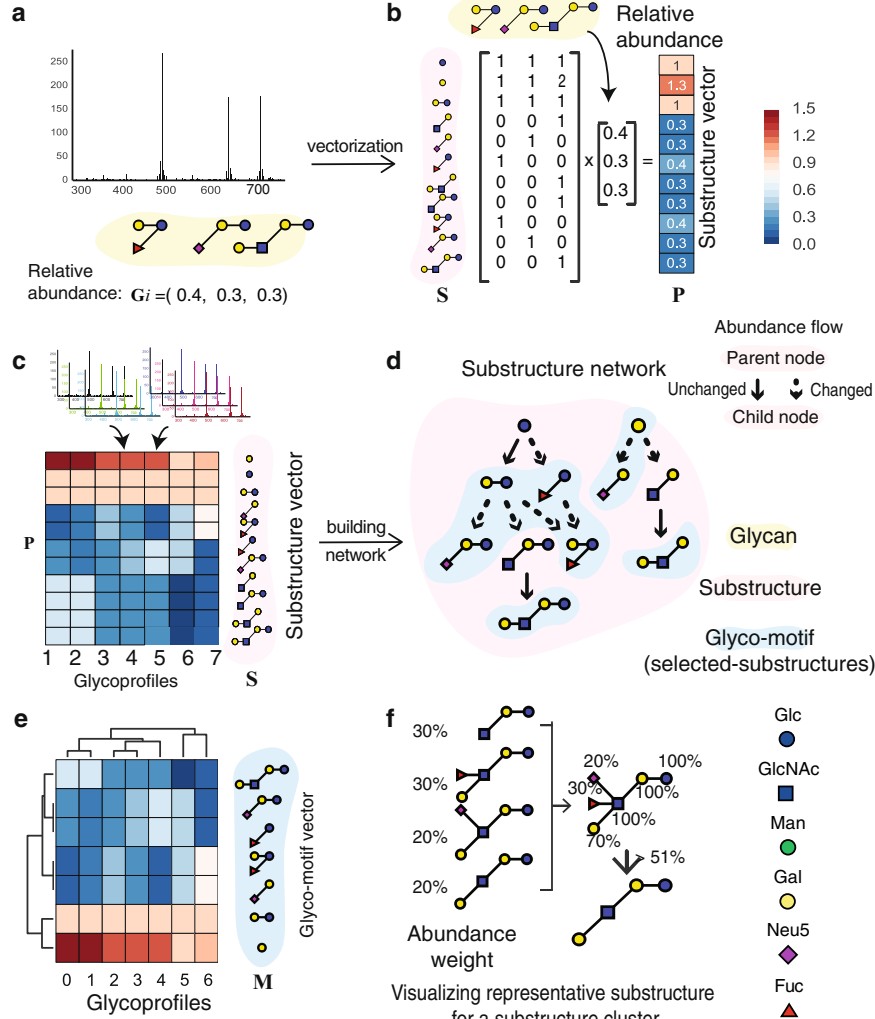

**Fig. 2 Core methodology for transforming glycoprofiles to glyco-motif profiles and visualizing cluster-representative substructures using GlyCompare.**
**a**, **b** A glycoprofile with structure and relative abundance annotation **G** is obtained. The glycans are decomposed to a substructure set **S**, and the presence/absence of each substructure is recorded. Presence/absence vectors are weighted by the glycan abundance, and are summed into a substructure vector **P**. **c** Seven example glycoprofiles are transformed to substructure vectors as (**a**) and (**b**). **d** A substructure network is constructed to identify the non-redundant glyco-motifs that change in abundance from their precursor substructures. **e** The glycoprofiles can then be compared by their glyco-motif vectors **M** to generate more meaningful clusters. Both glycoprofiles and substructures can be clustered for similarity analysis. **f** Core structure information can be visualized from a substructure cluster. For example, four substructures with different weights were aligned together, and the monosaccharides with a weight over 51% were preserved. Throughout the manuscript, glycan is referred to complete and secreted monosaccharide polymer; a glycan substructure is referred to a complete or incomplete monosaccharide polymer observable within at least one secreted glycan; a glycan motif (glyco-motif) is referred to an enriched functional glycan substructure for a dataset or biological process. Note that both glycan epitopes (typically terminal glycan substructures recognized by lectins) and glycan cores (biosynthetic glycan substructures common to select types (e.g., N- or O-glycosylation) or modes (e.g., complex or high-mannose) of biosynthesis) are glyco-motifs as they are biologically functional, interpretable and will be enriched in datasets selecting for specific glycan presentation of biosynthesis. Glycompare core methods are explained at length in the "Methods" section.

utilized, the clustering is heavily affected by the categorical presence or absence of a glycan, rather than structural similarity.

GlyCompare addresses these problems by exposing hidden similarities between glycans after decomposing glycoprofiles to their composite substructures. The 16 glycoprofiles with 52 glycans in total were decomposed into their constituent glycan substructures, resulting in a substructure network with 613 glycan substructures (Fig. 3b, c). Furthermore, the known enzymatic rules are annotated to the edges and the network is collapsed to include 151 glyco-motifs (Fig. 3c). By encoding the structure information, the glyco-motifs provide richer information than using glycans alone (Fig. 3d–f). The glyco-motif clustering clearly distinguished the samples based on the structural patterns and

separated profiles into groups more consistently than the raw glycan-based clusters (Fig. 3b and Supplementary Figs. 3–5).

The 16 glycoprofiles clustered into three groups with a few severely modified outliers (Fig. 3b). The 151 glyco-motifs were clustered into 35 groups, each summarized by representative substructures Rep1–Rep35 (Fig. 4a and Supplementary Fig. 1). The clusters of glycoprofiles are consistent with the genetic similarities among the host cells. Specifically, the major substructure patterns cluster individual samples into four categories: 'wild-type (WT)-like', 'mild', 'medium', and 'severe'. The WT-like category contains WT and B4galt1/2/3/4 knockouts samples, which have most substructures seen in WT cells. The mild group includes the Mgat4b/4a, Mgat4b, and Mgat5

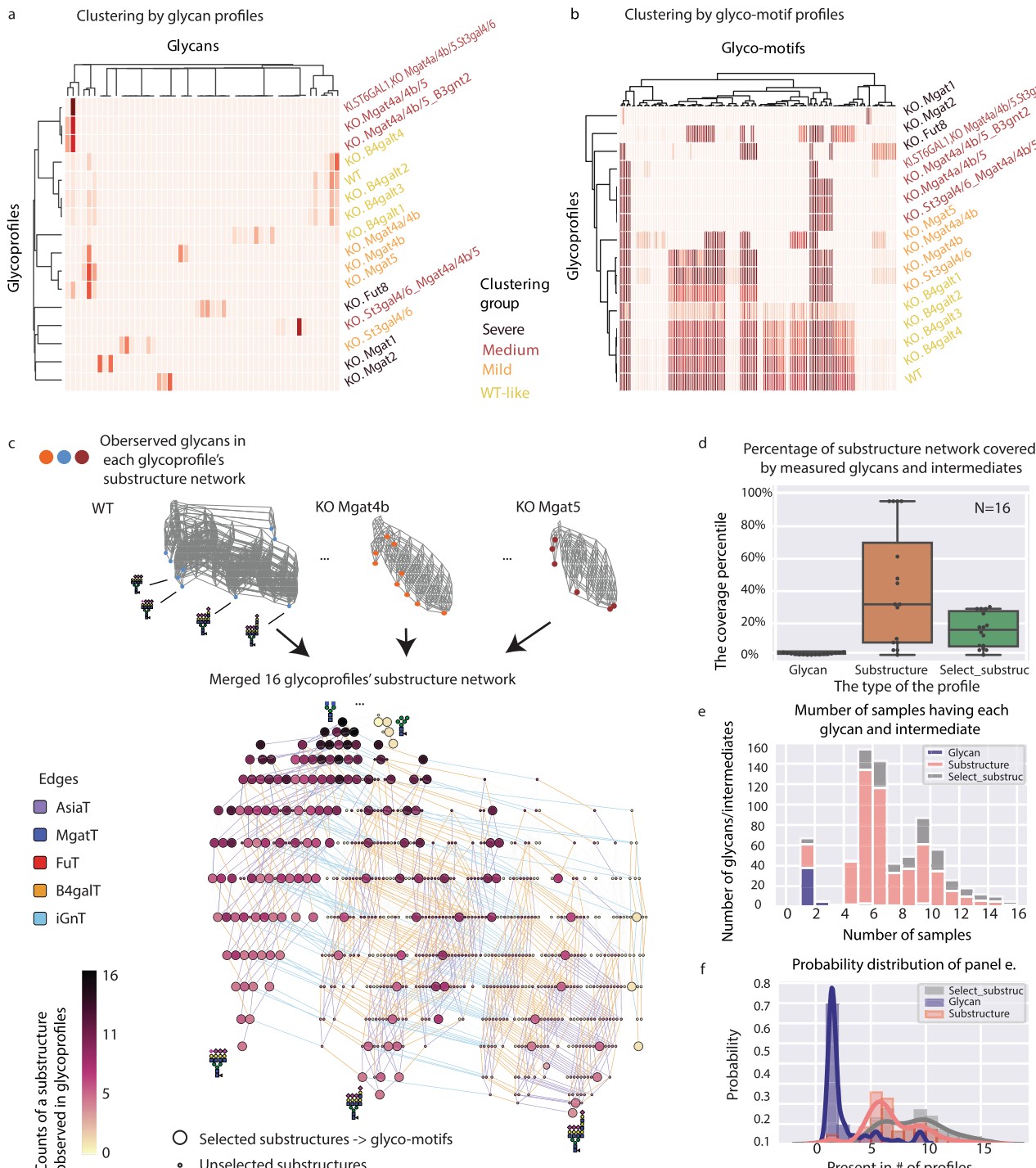

**Fig. 3 Substructure-based profile comparison solves the glycan non-independence and sparsity challenges, enabling the use of hierarchical clustering on large glycomics datasets. a** Clustering of unprocessed glycoprofiles. Sixteen glycoprofiles from glycoengineered recombinant EPO were clustered based solely on their raw glycan abundances. **b** Clustering of glyco-motif profiles. The glyco-motif profiles, constructed using GlyCompare, were clustered based on the 151 glyco-motifs (see "Methods"). There are four different phenotypic glycoprofiles (based on the glycoengineered glycosylation changes relative to wild type): WT-like (yellow), Mild (orange), Medium (red), and Severe (brown). The clusters of glycoprofiles and glycan substructures are defined by distance threshold = 0.5. In both cases, hierarchical clustering was used with a complete linkage and correlation-distance using seaborn 0.9.0. **c** The pan-network (516 intermediate substructures) that describes the synthesis of all glycans measured on the 16 glycoengineered recombinant EPO N-glycoprofiles. The glyco-motifs (in larger size) are the minimal set of 151 substructures selected by GlyCompare for further multi-glycoprofile comparison. The edges are colored by the enzyme family, AsiaT (purple), MgatT (blue), Fut (red), B4galT(orange), iGnt(cyan) and the node color according to frequency of existence in 16 glycoprofiles. **d** The coverage of the entire glycan synthetic pathway for 16 glycoprofiles using different structure types: glycan (purple, $n = 16$, Min = 0.00589, Q1 = 0.00786, median = 0.0128, Q3 = 0.0177, Max = 0.0236), substructure (gray, $n = 16$, Min = 0.005894, Q1 = 0.082515, median = 0.318271, Q3 = 0.698428, Max = 0.954813), and the selected-substructure (orange, $n = 16$, Min = 0.005894, Q1 = 0.058448, median = 0.161100, Q3 = 0.276031, Max = 0.290766). **e** Proportion of samples containing a glycan, substructure, or glyco-motif in the 16 samples, and **f** the associated probability distribution. Source data are provided as a Source data file.

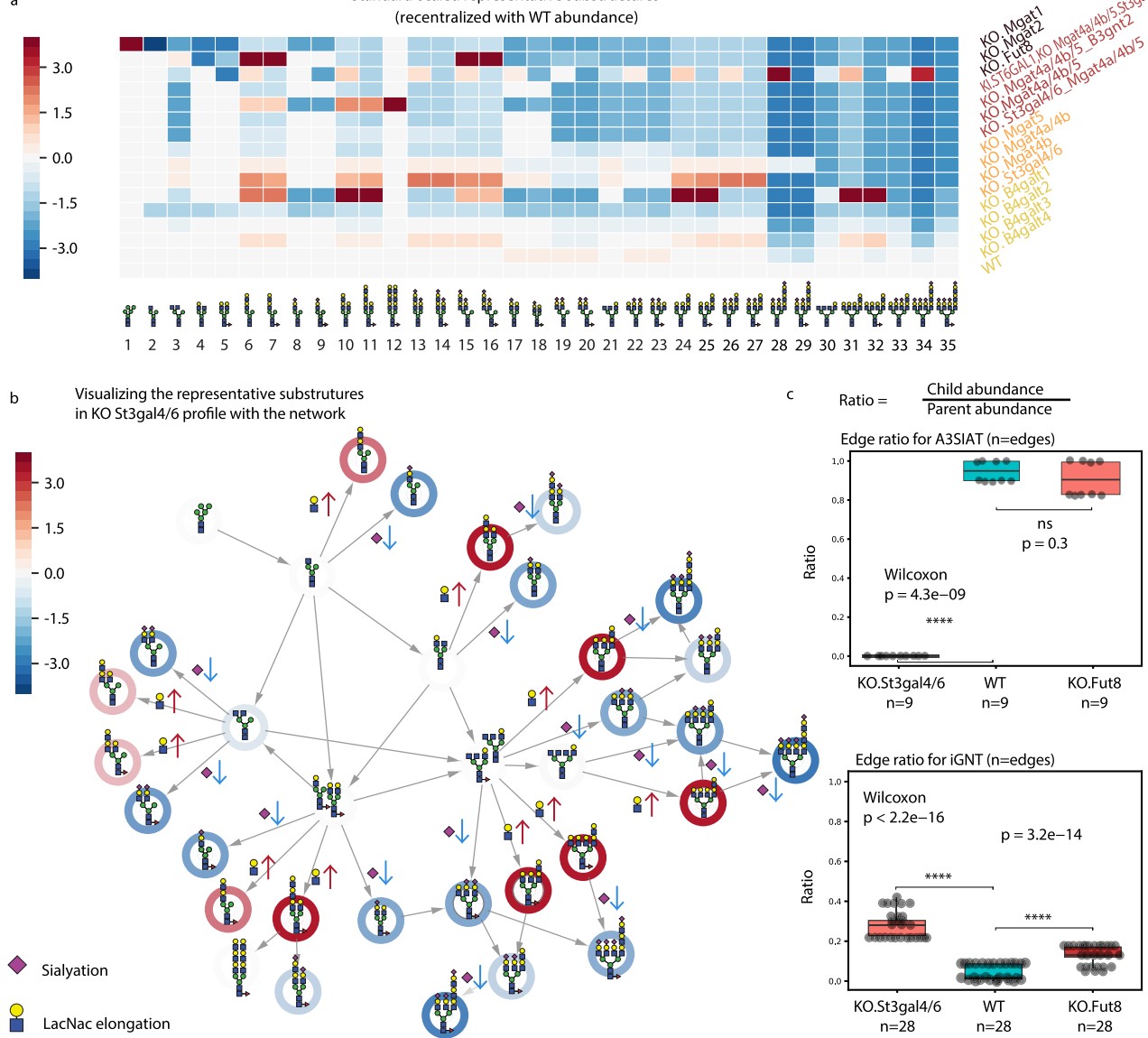

**Fig. 4 Analysis of glycan abundance changes using representative substructures. a** The heatmap of normalized glycan abundance for the 35 substructure clusters from Fig. 3b. The substructures are sorted based on the glycan structure complexity, followed by the number of branches, the degree of galactosylation, sialylation, and fucosylation. The profile colors are same as in Fig. 3b. While comparing to WT, the weighted average abundance of each cluster is calculated then z-score standardized by each column. The color denotes the change of glycan abundance for the comparison of KO vs. WT of the indicated substructure. **b** The differential substructure representative network for the comparison between the St3gal4/6 knockout profile and the WT profile. The z-score rescaled substructure clusters' abundance in (**a**) are visualized on edges with a simplified network. The color is defined the same as in (**a**) for the changes of glycan abundance. The plot demonstrates the changes of the elongation and sialylation. **c** Differential enzyme activities of α-2,3-sialyltransferase (a3SiaT, reaction $n = 9$) and β-1,3-N-acetylglucosaminyltransferase (iGNT, reaction $n = 28$) for the knockout profiles (St3gal4/6 and Fut8) and wild-type profile in terms of network edge ratio. Specifically, the network edge ratio is calculated on the reactions shared by three profiles. The 5 quartile boundaries of the a3SiaT table are KO.St3gal4/6, Min = 0, Max = 0;KO.Fut8, Min = 0.795, Q1 = 0.795, median = 0.795, Q3 = 1, Max = 1; WT, Min = 0.871, Q1 = 0.871, median = 0.871, Q3 = 1, Max = 1. The 5 quartile boundaries of the iGNT table are KO.St3gal4/6, Min = 0.224, Q1 = 0.224, median = 0.285, Q3 = 0.314, Max = 0.412;KO.Fut8, Min = 0.096, Q1 = 0.104, median = 0.161, Q3 = 0.205, Max = 0.205; WT, Min = 0.0569, Q1 = 0.0637, median = 0.0709, Q3 = 0.129, Max = 0.129. The one-sided Wilcox tests are performed. For a3SiaT table, KO.St3gal4/6 vs WT has $p = 4.3e−09$, KO.Fut8 vs WT has $p = 0.3$. For iGNT table, KO.St3gal4/6 vs WT has $p < 2.2e−16$, KO.Fut8 vs WT has $p = 3.2e−14$. Source data are provided as a Source data file.

knockouts, where each loses the tetra-antennary structure, and a St3gal4/6 knockout, which loses the terminal sialylation. The medium category is a group that contains knockouts of St3gal4/6 and Mgat4a/4b/5, knockouts of Mgat4a/4b/5 and B3gnt2, knockouts of Mgat4a/4a/5 with a knock-in of human ST6GAL1, and knockouts of Mgat4a/4b/5 and St3gal4/6. The medium

disruption category lost the tri-antennary structure. The 'severe' category includes three individual glycoprofiles with knockouts for Fut8, Mgat2, and Mgat1, each of which generates many glycans not detected in the WT-like, mild or medium categories. While some glyco-motif clusters can be seen in the glycoprofile clusters, there are important differences, and the glyco-motif

clusters provide more information and improved cluster stability (Fig. 4a, Supplementary Figs. 4, 5). These results demonstrate that standard methods are unfit to cluster glycan abundance from glycomics data in genetically diverse datasets; however, computing glyco-motif abundance accounts for the structural similarity of glycans between different glycoprofiles and allows one to use standard hierarchical clustering techniques reliably.

**GlyCompare summarizes structural changes across glycoprofiles.** GlyCompare helps to more robustly group samples by accounting for the biosynthetic and structural similarities of glycans. Further analysis of the representative structures provides detailed insights into which structural features vary the most across samples. To accomplish this, we rescaled the representative structure abundances and identified significant changes between mutant cells and WT (Fig. 4a, Supplementary Fig. 6). Analysis of the representative substructure network provides a more precise interpretation of the changes in the St3gal4/6 KO (Fig. 4b) and the Fut8 KO profiles (Supplementary Fig. 7). This interpretation highlights the specific structural features of glycans that are impacted when glycoengineering recombinant EPO.

In-depth analysis showed, as expected, in the Mgat1 knockout glycoprofile, only high-mannose N-glycans are seen. Also, in the Mgat2 knockout, the glycan substructure of bi-antennary on one mannose linkage significantly increases. The unique structure of bi-antennary LacNac elongated in the N-glycans emerges in the St3gal4/6 and Mgat4a/4b/5 knockouts. In the St3gal4/6 knockout profile, the abundance of structures with sialylation are zero, while the tetra-antennary and tri-antennary poly-LacNAc elongated N-glycan substructure without sialylation significantly increased (Rep24–25: $p = 1.3 \times 10^{-3}$, Rep31–32: $p = 2.3 \times 10^{-4}$) (Fig. 4a–c). Along with expected changes in α-1,6 fucosylation in the Fut8 knockout glycoprofile, we also observed an increase in the tetra-antennary poly-LacNac elongated N-glycan without fucose, which has not been previously reported (one-sided one-sample Wilcoxon test, Rep28: $p = 2.7 \times 10^{-4}$, Rep34: $p = 2.0 \times 10^{-4}$) (Fig. 4a). Both the St3gal4/6 and Fut8 knockout profiles have increased tri/tetra-antennary poly-LacNac elongated substructure (Rep24, Rep31). It is related to the increased conversion ratio of iGNT (Fig. 4c). Finally, the Mgat4b, Mgat4a/4b, and Mgat5 knockouts lose all core tetra-antennary substructures (Rep30–35: unscaled abundance = 0) (Supplementary Fig. 6). While tri-antennary substructures with GlcNac elongation increased significantly for Mgat4b (Rep13–14, $p = 2.6 \times 10^{-3}$; Rep26–27: $p = 2.5 \times 10^{-4}$), the poly-LacNac elongation structure disappeared. Interestingly, while both the Mgat4b and Mgat5 knockouts do not have the tri-antennary poly-LacNac elongated N-glycan, the Mgat4a/4b mutant keeps a highly abundant poly-LacNac branch (Rep28–29: $p = 2.4 \times 10^{-4}$). Thus, by using GlyCompare, we identified the specific glycan features impacted not only in individual glycoengineered cell lines but also in features shared by groups of related cell lines.

**GlyCompare reveals unexpected changes in substructures invisible at the whole-glycan level.** Many secreted and measured glycans are also precursors, or substructures, of larger glycans (Fig. 5a). Thus, the secreted and observed abundance of one glycan may not equal the total amount synthesized. GlyCompare quantifies the total abundance of a glycan by combining the glycan abundance with the abundance of its products. To demonstrate this capability of GlyCompare, we analyzed HMO abundance, to test if maternal genetics underlying the secretor status has unexpected off-target effects on other HMO features. We obtained 47 HMO glycoprofiles from 6 mothers (1, 2, 3, 4, 7, 14, 28, and 42 days postpartum (DPP)), 4 "secretor" mothers with

functioning FUT2 (α-1,2 fucosyltransferase), and 2 "non-secretor" mothers with non-functional FUT2. With GlyCompare addressing the interdependence of HMOs, we could use powerful statistical methods to study trends in HMO synthesis. Specifically, we used regression models to predict secretor status and DPP from substructure abundance.

We first checked both the glycan-level and substructure-level clustering of the glycoprofile. Samples with same secretor status and days postpartum (DPP) were successfully grouped (Supplementary Figs. 8, 9). Further examination of the glyco-motif abundance (i.e., the total amount of substructure synthesized) revealed phenotype-related trends invisible on the glycan profile level. Interestingly, secretor status, defined by glycan fucosylation, significantly impacts the sialylation of non-fucosylated HMOs (e.g. LSTb) over time. While the relative abundance of both LSTb substructure (X62) and secreted LSTb was elevated in non-secretor milk (Wald $p = 4 \times 10^{-7}$ and Wald $p = 3.98 \times 10^{-13}$; Fig. 5a, b), only X62 showed a strong interaction between time and secretor status. At an adjusted sample size of 6, the time-dependent decrease in non-secretor X62 is significant (Wald $p = 0.002$). In contrast, the time-dependent decrease is only marginally significant for secreted LSTb (Wald $p = 0.03$). Previous work has already described an LSTb elevation at 3–4 months postpartum[36]. Here, a substructure-analysis of X62 suggests that while the secreted LSTb is elevated in non-secretor milk, total LSTb produced (and consumed as the substrate for other sugars) may decrease over time.

Examining other secreted HMOs containing the X62 substructure (DSLNT and DSLNH), we see no significant secretor-status-dependent elevation (Wald $p > 0.2$; Fig. 5a–d). Unlike X62, DSLNT shows no significant change over time (Coef $= -0.39$, Wald $p = 0.17$; Supplementary Table 3a). Finally, DSLNH shows a significant increase over time (Wald $p = 2.91 \times 10^{-8}$; Supplementary Table 3a). The secretor-specific trends in total LSTb are only clearly visible by examining the X62 substructure abundance (Fig. 5a–d). Thus, while secretor status is expected to impact HMO fucosylation, GlyCompare reveals associations with non-fucosylated substructures. Viewing substructure abundance as total substructure synthesized provides a fundamental measure to the study of glycoprofiles (Supplementary Fig. 10); it also creates an opportunity to explore trends in biosynthesis.

**Flux estimation from GlyCompare identifies reaction responsible for an unexpected change in sialylation.** The identification of a non-fucosylated substructure that is associated with differences in secretor genotype raised the question of which reactions are responsible. Thus, we used GlyCompare to estimate enzyme fluxes to identify the reaction responsible for the unexpected change in HMOs. To do this, we estimate the flux for each biosynthetic reaction by quantifying the abundance ratio of products and substrates from parent–child pairs of glycan substructures. Thus, we could study changes in HMO synthesis through the systematic estimation of reaction flux across various conditions.

Although the fucosyltransferase-2 genotype defines secretor status, not all secretor-associated reactions were fucosylation reactions. We further explored the secretor-X62 association using the product-substrate ratio to estimate flux. Specifically, we examined the upstream reaction of LNT (X40) to LSTb (X62) and the downstream reaction of LSTb (X62) to DSLNT (X106) (Fig. 5e). We estimated the flux of the upstream reaction of LNT converting to LSTb, using the X62/X40 ratio over time. However, no significant change was observed to secretor status (Fig. 5f; Wald $p = 0.55$). In the conversion of LSTb to DSLNT, we found a secretor-specific increase in reaction flux. Specifically, the X106/

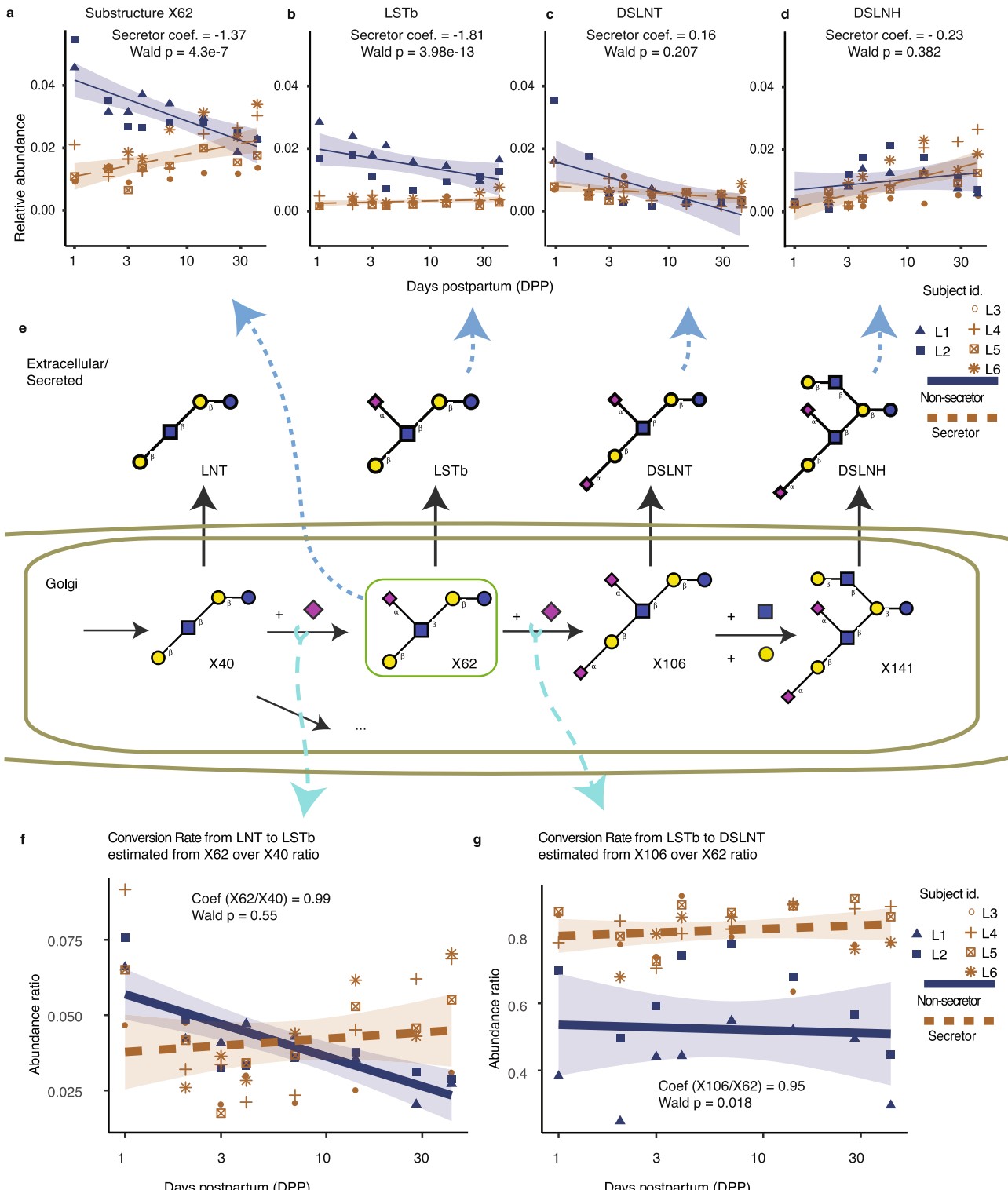

X62 ratio was significantly higher (Wald $p = 0.018$) in secretor mothers (Fig. 5g; Supplementary Table 3c). In the average non-secretor mother, 52.3% (s.d. 15.1%) of LSTb is converted to DSLNT. Meanwhile, in secretors, on average, 81.8% (s.d. 7.2%) is converted. The LSTb to DSLNT conversion rate appears higher in secretors, while conversion from the LSTb precursor, LNT, appears unchanged (Fig. 5f). Any changes in sialylation are intriguing, considering that secretor status is associated with genetic variation of a fucosyltransferase. A secretor-elevated conversion rate from LSTb to DSLNT is consistent with

observing elevated X62 and secreted LSTb in non-secretor milk (Fig. 5a, b)[36]; if non-secretors consume less LSTb as a DSLNT substrate, more of the synthesized LSTb (X62) will remain LSTb through secretion. Examining the product-substrate ratio has revealed a phenotype-specific reaction propensity, thus providing insight into the condition-specific synthesis.

**GlyCompare increases the statistical power of glycomics data.** GlyCompare successfully provides insights by accounting for

**Fig. 5 Analysis of intermediate substructures with GlyCompare elucidates unexpected associations in HMO abundance and reaction flux with secretor status, which are missed in the standard whole-glycan analysis.** **a**–**d** Over time (DPP), substructure X62, LSTb, DSLNT, and DSLNH show different trends for secretors and non-secretors. Furthermore, the abundance of aggregated X62 shows significant positive-correlation with secretor and negative-correlation with non-secretor. GEE models for each structure are visualized and approximated using a gaussian-link generalized linear model with 95% confidence intervals; odds ratio (OR) significance (likelihood OR is non-zero) was measured with a two-sided Wald test (**a** $n = 47$, Coef $= -1.37$, $p = 4.3e-7$; **b** $n = 47$, Coef $= -1.81$, $p = 3.98e-13$; **c** $n = 47$, Coef $= 0.16$, $p = 3.98e-13$; **d** $n = 47$, Coef $= 0.382$, $p = -0.23$). **e** The substructure intermediates for four connected glycans are shown here. The synthesis of larger glycans must pass through intermediate substructures that are also observed glycans, where the substructures are as associated with measured glycans as follow X40 = LNT, X62 = LSTb, X106 = DSLNT, X138 = DSLNH. **f**, **g** Panels examine the product-substrate ratio for two reactions in panel (**e**). X40, the LNT substructure, is a precursor to X62, the LSTb substructure, which is a precursor to X106, the DSLNT substructure. We estimate the flux of these conversions from X40 to X62 and X62 to X106 by examining the product-substrate ratio, i.e., the proportion of the total synthesized substrate converted to the product. LSTb/LNT substructure relative abundance ratios are not associated with secretor status while DSLNT/LSTb ratios are. Panels **f** and **g** show OR corresponding to the ratio association with secretor status (**f** $n = 47$, OR $= 0.99$, $p = 0.55$; **g** $n = 47$, OR $= 0.95$, $p = 0.018$). See Supplementary Table 3 for complete GEE statistics. Source data are provided as a Source data file.

shared biosynthetic routes of measured oligosaccharides. Since it includes information on the similarities between different glycans, we wondered how our approach impacts statistical power in glycan analysis. Thus, to quantify the benefit of the glyco-motif analysis, we constructed many regression models associating either glyco-motif abundance or glycan abundance, with a DPP and secretor status (see "Methods"). We found that regressions trained with glyco-motif abundance are more robust than those trained on whole glycan abundance, as indicated by the increased coefficient magnitude (Wilcoxon $p = 0.0047$, Fig. 6a) and decreased standard error (Wilcoxon $p = 0.033$, Fig. 6b). An increase in the stability of a statistic can result in an increased effect size. Consistent with the increased coefficient magnitude and decreased standard error, the effect size also increased, as measured by the marginal $R^2$ ($mR^2$) of glyco-motif-trained regressions (Wilcoxon $p = 0.04$, Fig. 6c). These effects were confirmed with a bootstrapping $t$-test; bootstrapping $p$-values were less than or equal to Wilcoxon $p$-values within 0.001. Increases in statistical magnitude, statistical stability, and effect size are all expected to increase analysis power. Using the median, 1st quartile, and 3rd quartile of observed $mR^2$, we estimated the expected power of glyco-motif-trained and glycan-trained regressions at various sample sizes. The expected power of a glyco-motif-trained regression reaches 0.8 at 36 samples and 0.9 at 57 samples. In contrast, a glycan-trained regression requires more than double the sample size to reach a comparable power (Fig. 6d). GlyCompare provides additional power for discovering glycan-phenotype associations.

To further probe the increased statistical power, we compared our approach to another statistically-driven network approach. Benedetti et al.[28] demonstrated that novel glycan biosynthetic reactions could be resolved using partial correlation[28]. Using the Benedetti data, we computed partial correlation for glycan abundance and with GlyCompare-computed linkage-specified substructure abundance. We compared the partial correlation between glycans or substructures across true-positive, known reactions and false-positive, uncharacterized reactions (as specified in the Benedetti supplement). Partial correlations across known reactions between GlyCompare-computed substructures were significantly higher than partial correlations between corresponding glycan abundances (Supplementary Fig. 11). Partial correlation across known reactions was elevated for substructure abundance in all IgG isoforms (one-sided $t$-test, $p < 0.0039$), and reactions performed by B4GALT1 and ST6GAL1 (one-sided $t$-test, $p < 1.1 \times 10^{-4}$). Interestingly, the lowest partial correlations across true-positive reactions between substructures were substantially higher than corresponding glycan correlations. The higher floor for substructure correlations suggests that substructure abundances may increase positive predictive value (Supplementary Fig. 11). Finally, while correlation increased

between true-positive associated substructures, correlations across uncharacterized reactions were close to zero and indistinct from glycan correlations across the same reactions. Thus, using GlyCompare for glyco-motif-level analysis can substantially increase the robustness and statistical power in glycomics data analysis since it allows for comparing different glycans who share biosynthetic steps.

**Additional statistical power reveals tumor-depleted mucin-type O-glycans.** To explore the broad applicability of GlyCompare, we used our method to calculate substructure abundance for mucin-type O-glycans[37] (Fig. 7), glycolipids[38] (Supplementary Fig. 12), and site-specific compositional N-glycosylation[22] (Supplementary Fig. 13). These results are described in more detail in the Supplementary Discussion.

In a re-examination of the mucin-type O-glycans from tumor and normal samples, glycan abundance and motif abundance were compared (Fig. 7a, b). We found zero whole-glycan structures significantly distinguished between tumor and normal following multiple-test correction (FDR < 0.1, Fig. 7a). Yet, after substructure decomposition using Glycompare, we found five significantly depleted (FDR < 0.1) mucin-type glycan motifs in gastrointestinal cancer (Fig. 7b)[37]. We found a substantial depletion in the tumor samples of five core 2 structures. These structures included three fucosylated and two with I-branches. The largest structures were over 30-fold depleted in tumors (FDR < 0.03, Fig. 7c). The core 2 depletion was noted as a nonsignificant trend in the original publication; we identified the specific core 2-type substructure depleted in tumors using substructure decomposition. Though this dataset contains few subjects and therefore may not be robustly generalizable, we demonstrate the increase in statistical power when using substructures. In addition, a later study also found significant depletion of multiple bi-GlcNAc core 2 and I-branched structures[37]. Also consistent with the decrease in bi-GlcNAc core-2 structures in gastric cancer, low expression of B3GNT3 in stomach cancer is significantly associated with decreased overall survival[38]. B3GNT3 is necessary for adding the second GlcNac to core 2 structure[39] and therefore upstream of all significantly depleted structures (Fig. 7); B3GNT3 depletion could explain the observed differential glycosylation. The observation of significantly distinct substructures suggests GlyCompare provided increased statistical power to detect these distinguishing condition-enriched structures, and further showed continuity across similar structures was not evident in the original study.

## Discussion
Glycosylation has generally been studied from the whole-glycan perspective using mass spectrometry and other analytical methods. From this perspective, two glycans that differ by only one

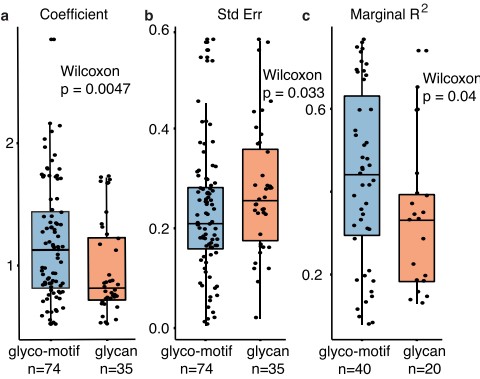

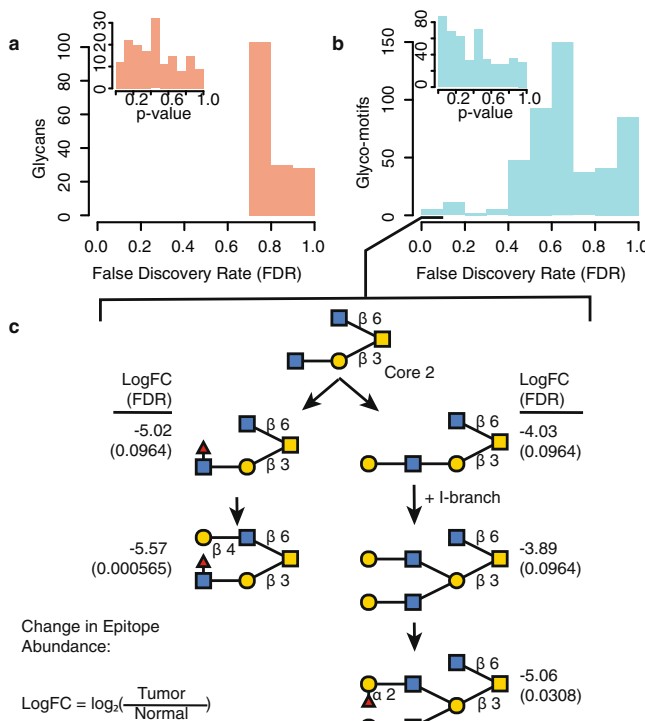

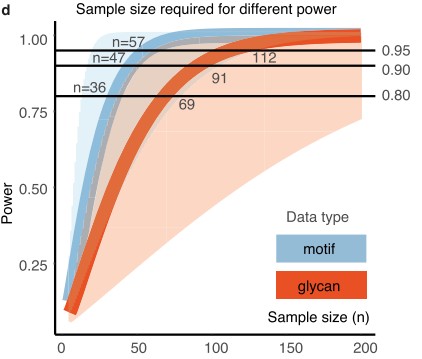

**Fig. 6 Glyco-motif level statistics require half as many samples to reach the same level of statistical power as analysis with raw glycans. a, b** The use of glyco-motifs improves measures of regression robustness. The coefficient magnitude and Standard Error indicate the magnitude of the measured effect and the confidence with which a coefficient can be estimated. In **a**, the boxplot illustrates 25th, 50th, and 75th percentiles for regression coefficients using glycan data (Min = 0.5094, Q1 = 0.7206, median = 0.8416, Q3 = 1.2706, Max = 1.7166, $n$ = 35) or glyco-motif data (Min = 0.5094, Q1 = 0.8365, median = 1.1403, Q3 = 1.5106, Max = 2.8357, $n$ = 74). Distributions were compared using one-sided Wilcoxon tests ($p$ = 0.0047). In **b**, the boxplot again illustrates the 25th, 50th, and 75th percentiles for regression standard error trained on glycan data (Min = 0.0182, Q1 = 0.1631, median = 0.2446, Q3 = 0.2832, Max = 0.4518, $n$ = 35) or glyco-motif data (Min = 0.0053, Q1 = 0.1508, median = 0.2047, Q3 = 0.2747, Max = 0.5398, $n$ = 74). Distributions were compared using a one-sided Wilcoxon test ($p$ = 0.033). **c** The $R^2$ describes the effect size of a regression; we used marginal $R^2$ (m$R^2$) because it was appropriate for the regression models used[51]. Distributions for m$R^2$ of regression models trained on glycan data (Min = 0.128, Q1 = 0.183, median = 0.331, Q3 = 0.441, Max = 0.737, $n$ = 20) and glyco-motif data (Min = 0.0949, Q1 = 0.3185, median = 0.46, Q3 = 0.686, Max = 0.764, $n$ = 40) were compared using a one-sided Wilcoxon test ($p$ = 0.04). **d** We predicted power for a range of sample sizes ($n$ = 5–200) given the median effect size (solid line) within the interquartile range (shaded region) for glyco-motif-trained regressions (m$R^2$: Q1 = 0.31, median = 0.45, Q3 = 0.68) and the median effect size for glycan-trained regressions (m$R^2$: Q1 = 0.18, median = 0.33, Q3 = 0.44). Here, the use of GlyCompare and glyco-motif (grey-blue color) abundances required approximately half the number of samples to achieve equivalent power as standard glycan (red color) measures. Source data are provided as a Source data file.

**Fig. 7 Increased power for identifying diagnostic markers shown through a re-analysis of mucin-type O-glycans from normal, tumor-proximal, and gastrointestinal cancer biopsies, transformed to motif abundance. a, b** Welch two-sample $t$-test $P$-value and false discovery rate (FDR) distributions for glycan abundance and glyco-motif abundance. **c** We found multiple core 2 substructures depleted in gastrointestinal cancer relative to normal tissue. Not all linkages are specified, only those relevant to the substructure definition. The information of log fold changes (logFC) and the FDR are presented next to each substructure. Source data are provided as a Source data file.

Glycoprofiles are converted to glyco-motif profiles; wherein each substructure abundance represents the cumulative abundance of all glycans containing that substructure. In other words, substructure abundance is automatically computed from given glycoprofiles. Motif selection highlights the minimal set of substructures necessary to understand variation in the given glycoprofiles. Our substructure quantification can be easily scaled up to compare many glycoprofiles in large datasets. Thus, it brings several advantages and different perspectives, with some important limitations, to enable the systematic study of glycomics data.

Like any analytical pipeline, GlyCompare is sensitive to upstream analysis (e.g., mass spectrometry methods measure the mass-to-charge ratios of glycans and their fragments, and thus require expert annotation to assign structures). Therefore, GlyCompare will continue to improve with advances in glycoprofile structure annotation quality. Going forward, we hope to include multiple methods for aggregating abundance over substructures, including aggregation using multiple functions (besides addition) over fully or partially specified biosynthetic networks. While summing abundance for all subsumed substructures works well, manual reaction specification can help avoid information loss when biosynthesis is not hierarchical and acyclic or glycans are not increasing in size. When these limitations are acknowledged, the current version of glycompare has demonstrated some exciting capabilities.

First, the GlyCompare platform computes a glyco-motif profile (i.e., the abundances of the minimal set of glycan substructures)

monosaccharide are distinct and are not directly comparable. Thus, the comparative study of glycoprofiles has been limited to changes between glycans shared by multiple glycoprofiles or small manually curated glycan substructures[29]. GlyCompare sheds light on the hidden biosynthetic relationships between glycans by integrating the structural similarity into the comparison.

that maintains the information of the original glycoprofiles, while exposing the shared intermediates of measured glycans. These glyco-motif profiles more accurately quantify similarities across glycoprofiles. This is made possible since glycans that share substructures also share many biosynthetic steps. If the glycan biosynthetic network is perturbed, any synthesized glycan can be impacted and the nearest substructures can directly highlight where the change occurred. For example, in EPO glycoprofiles studied here, the tetra-antennary structure is depleted in the Mgat4a/4b/5 knockout group and the downstream sialylated substructure depleted when St3gal4/6 were knocked out. Such structural patterns emerge in GlyCompare since the tool leverages shared intermediate substructures for clustering, thus identifying common features across diverse samples.

Second, trends in glycan biosynthetic flux become visible at the substructure level. For example, in the HMO dataset, multiple glycans are made through a series of steps from LNT to DSLNH (Fig. 5a). Only when the substructure abundances and product-substrate ratios are computed can we observe the secretor-dependent temporal differences in the abundance of the LSTb substructure, X62. Interestingly, though changes in α-1,2 fucosylation define secretor status, we see additional secretor-dependent effect on sialylated structures with no fucose. The biosynthetic interpretation of lactose-based substructures was also applied to ocular gangliosides[38] to identify tissue-specific glycolipid substructures (Supplementary Fig. 12). These are the systemic effects invisible without a systems-level perspective due to the interconnected nature of glycan synthesis; this disparity underlines the power of this method.

Third, the sparse nature of glycomic datasets and the synthetic connections between glycans make glycomic data unfit for many common statistical analyses. However, the translation of glycoprofiles into substructure abundance provides a framework for a more statistically powerful and robust analysis of glycomic datasets. These methods can enrich both structural (Fig. 3a) and compositional (Supplementary Fig. 13) thereby increasing the interpretability and structure of the dataset. Single sample perturbations, such as the knockouts in the glycoengineered EPO, can be compared to wild-type; all substructure data can be normalized and rigorously distinguished from the control using a one-sample Wilcoxon test. Furthermore, conditions or phenotypes with many glycoprofiles, such as the secretor status in the HMO dataset, can be compared using various statistical methods to evaluate the association between the phenotypes and glycosylation. For example, in HMO data, we confirmed that the α-1,2 fucose substructure is enriched in secretor status, consistent with previous studies[39–41]. Because the substructure approach includes comparisons of glycans that are not shared across the different samples but share intermediates, GlyCompare decreased sparsity and increased statistical power. We demonstrate the increase in statistical power and observable differences between HMO (Fig. 6) and the tumor-proximal mucin-type O-glycan presentation (Fig. 7). Thus, one can obtain richer glycan comparisons of representative substructures, total synthesized abundance, and flux.

Finally, in combination with the substructure network, we can systematically study glycan synthesis. The product-substrate ratio provides an estimation of flux through the glycan biosynthetic pathways. Using the HMO dataset, we demonstrated the power of this perspective by showing that more LSTb is converted to DSLNT in the secretor mother. The perspectives made available through GlyCompare are not limited by the independence assumptions of most statistical tests. Because the substructure-level perspective makes explicit biosynthetic dependency between glycans, glyco-motif abundances can be used with nearly any statistical model or comparison demanded by a dataset. We have accommodated the sparse and non-independent nature of glycoprofiles, thereby making countless comparisons analyses possible.

## Methods

**Overview of the pipelines**. Supplementary Fig. 14 shows a summary of the GlyCompare workflow. The GlyCompare workflow consists of several steps wherein glycoprofiles are annotated and decomposed, glyco-motifs are prioritized, and each glyco-motif is quantified for subsequent comparisons with or without specific phenotype data.

**N-glycosylation of EPO glycoprofile collection for re-analysis**. N-glycosylation data were previously published[27]. Upon retrieving these data from the study, we picked 16 glycoprofiles that are used again in their follow-up study[21] and further processed the data as follows. All measurements were taken from distinct samples.

Glycan substructures were extracted from the observed glycans. Substructure abundance was calculated from the glycan abundance of all glycans containing the substructure. The substructure network identifies a minimal set of 151 glyco-motif substructures to compare the mutants. Finally, representative substructures were extracted to pool abundance and summarize the structural changes across mutants. Each of these operations is further specified below.

**HMO glycoprofile collection and analysis**. HMOs were analyzed as de-identified samples previously for an independent study[35,42] at Baylor College of Medicine. Following Institutional Review Board approval (Baylor College of Medicine, Houston, TX), lactating women provided written informed consent. Women with diabetes or impaired glucose tolerance, anemia, or renal or hepatic dysfunction were excluded from the study. Women were 18–35 years of age, had uncomplicated singleton pregnancies with vaginal delivery at term (>37 weeks) and pregnancy body mass index (BMI) remained <26 kg m⁻². Infants were healthy and exclusively breastfed. Forty-eight milk samples were collected from 6 human mothers (1, 2, 3, 4, 7, 14, 28, and 42 days postpartum (DPP)). More information on subject selection, exclusion, study design, and breast milk collection has already been published[35,42].

Glycan composition and abundance were measured by high-performance liquid chromatography (HPLC) following fluorescent derivatization with 2-aminobenzamide (2AB, CID: 6942)[43,44]. Raffinose (CHEBI:16634, CID:439242), a non-HMO oligosaccharide, was added to each milk sample as an internal standard at the beginning of sample preparation to allow for absolute quantification. Of the 300–500 predicted HMO, the 16 most abundant HMO were detected based on retention time comparison with commercial standard oligosaccharides and mass spectrometry analysis, including 2-fucosyllactose (2′FL), 3-fucosyllactose (3′FL), 3-sialyllactose (3′SL), lacto-N-tetraose (LNT), lacto-N-neotetraose (LNnT), lacto-N-fucopentaose (LNFP1, LNFP2, and LNFP3), sialyl-LNT (LSTb and LSTc), difucosyl-LNT (DFLNT), disialyllacto-N-tetraose (DSLNT), fucosyl-lacto-N-hexaose (FLNH), difucosyl-lacto-N-hexaose (DFLNH), fucosyl-disialyl-lacto-N-hexaose (FDSLNH), and disialyl-lacto-N-hexaose (DSLNH). GlyTouCan IDs for each glycan are listed in Supplementary Table 2.

HMO measurements by HPLC were quantified using Chromeleon 7.2[45]. Technicians were blinded to metadata associated with each sample. One sample was excluded; the HPLC failed to quantify HMO in the day 1 sample collected from subject L6, therefore, no data from this sample could be included. Samples were analyzed in a random order to mitigate batch effects. In addition to absolute concentration of each glycan $g_i$, the proportion of each glycan per total glycan concentration (sum of all integrated glycans) was calculated and expressed as relative abundance (% of the total, $g_i/\Sigma g_*$). The presence of 2-FL defines secretor status. Absolute abundance of HMO is determined by a well-characterized low-noise method[43,44] using HPLC analysis[46]. Therefore, no technical replicates were necessary.

HMO abundance profiles were treated similarly to the N-glycans. We identified and quantified 26 glyco-motifs from 121 substructures. We compared glyco-motif abundance and their abundance ratios directly to secretor status along with the log of days postpartum.

**Computing glycan substructure profiles from glycoprofiles**. Three procedures were used for pre-processing the studied glycoprofiles (Fig. 2a, b). First, glycoprofiles are parsed into glycans with abundance. In each glycoprofile, the glycans are manually drawn and exported with GlycoCT format using the GlyTouCan Graphic Input tool[23]. GlycoCT formatted glycans are loaded into Python (version 3+) and initialized as glypy.glycan objects using the Glypy (version 0.12.1)[47]. Assuming we have a glycoprofile $i$, the corresponding abundance of each glycan $j$ in glycoprofile $i$ is represented by $g_{ij}$. For example, the relative m/z peak in the mass spectrum or the abundance value in an HPLC trace, is calculated relative to the total abundance of glycans in this glycoprofile $g_{ij}/\Sigma g_{i*}$. Glycans with ambiguous topologies are handled by assuming they belong to every possible structure with equal probability, thereby creating all possible $n$ structures, still, with $g_{ij}/n\Sigma g_{i*}$ abundance of each. Second, glycans are annotated with glycan substructure information, and this information is transformed into the substructure vector.

Substructures within a glycan are exhaustively extracted by breaking down each linkage or a combination of linkages of the studied glycan. Note that this method cannot currently deal with cyclic glycans. All substructures extracted are merged into a substructure set **S**. Substructures are sorted by the number of monosaccharides and duplicates are removed. Then, each glycan is matched to the substructure set **S**, producing a binary glycan substructure presence (1) or absence (0) vector, $x_{ij}$. Last, a substructure (abundance) vector is calculated as $\mathbf{P}_i = \Sigma x_{ij} g_{ij} / \Sigma g_{i*}$ representing the abundance of the substructures $s$ in this glycoprofile, where $\mathbf{P}_i = (s_{1i}, ..., s_{ni})$. Third, a substructure network is built based on the substructure vectors. The substructure network is a directed acyclic graph wherein each node denotes a glycan substructure. Given the substructure set **S**, the root node starts from the monosaccharides or a defined root core structure, and a child node is a substructure with only one monosaccharide added to its parent node. We note that one child node might have multiple parent nodes and vice versa. The child node depends on its parent node(s) since it cannot exist without any parent node. The edges in the substructure network were annotated with known biosynthetic rules for further analysis. Substructure networks were visualized by networkx (version 2.1; https://networkx.org/). and cytoscape (version 3.8.2)[48].

**Selecting glyco-motifs from the substructure network**. A larger subset of the substructure network is necessary to uniquely describe a more diverse set of glycoprofiles, while fewer substructures are needed to describe more similar glycoprofiles sufficiently. Comparisons become more focused when only examining these variable substructures. To simplify the substructure network, the parent/child substructure pair that have the same abundance can be merged without any information loss. As illustrated in Fig. 2d, a parent–child substructure pair with the same abundance (solid arrow) can be merged. If they have the same abundance, we can conclude that the addition of the specific monosaccharide is not perturbed across all glycoprofiles, which means they carry the same information. Thus, the parent node can be pruned without information loss. All remaining nodes, namely, the glyco-motifs, are used to cluster the glycoprofiles.

After selecting the glyco-motifs (Fig. 2d), we use the "monosaccharides weight" to track whose parent node is merged. All node weights are initialized as 1. When a node is removed, the weight is equally divided and distributed to child nodes that have the same abundance as the removed node. Because the method redistributes weight from the root to leaves, descendant substructure nodes with differential abundance (relative to their parent node(s)) will gain additional weight. The weights **W** are used later for generating the representative substructures.

**Substructure-based clustering of glycoprofiles**. After generating the glyco-motifs, the Pearson correlation and 'complete' distance are used to cluster the glycoprofiles and substructures (Fig. 2e). The elbow method is used to determine the cluster numbers.

To identify the representative glycan substructures, a set of glycan substructures with weights **W** is first aligned (Fig. 2f). Then, we calculate the sum of monosaccharide weights for each glycan substructure. The representative substructure is thus defined as the glycan substructures with their summed monosaccharide weights >51% (a heuristic and flexible parameter to facilitate user-controlled clarity) of the total weight of glycan substructures. Last, the averaged abundances of the representative substructures are generated. Differential abundance of representative substructure can then be compared across glycoprofiles.

**Substructure cluster abundance comparison and network edge-based ratio comparison**. We use the representative substructures to summarize and analyze the structural and quantitative changes across glycoprofiles. For the abundance of a representative substructure in a glyco-motif cluster, we combine the substructure abundance and the substructure monosaccharide weights to generate the weighted average of substructure abundance. Since the abundance range of representative substructures across different glycoprofiles is different, we re-centralize the representative substructure abundance based on WT and scaled them with standard deviation. There are many representative substructures significantly deviating from the WT's abundance. Since the abundance distributions are not normally distributed, we used a one-sided 1-sample Wilcoxon test to test if the abundance of a representative substructure in a glycoprofile is significantly divergent. Effect size, $r$, was calculated as $\frac{z}{\sqrt{N}}$[49]. A Bonferroni correction ($n = 16$) was used to correct for multiple testing, so $p = 0.0031$ is used as criteria, and effect sizes are all above 0.68.

For those network edges annotated with enzyme information, we further test if an enzyme has the same efficacy in two glycoprofiles. Every edge has a parent/child abundance ratio. All edges annotated with the same enzyme consist of an abundance ratio distribution in one glycoprofile (Fig. 3c). The Wilcoxon test is used to compare the ratio distribution for the same enzyme in two glycoprofiles.

To have a concise view of the representative substructure network, we further generate a simplified network. The nodes from the substructure network are merged based on the substructure clustering. The edges connecting the original nodes are merged to connect the new nodes. Last, the derived representative substructure network represents the merged nodes and the edges annotated by enzymatic rules (Fig. 4b).

**Phenotype-associated substructure detection**. For revealing the phenotype-associated substructures, we estimated the influence of secretor status on glycan and glyco-motif abundance using a generalized estimating equation (GEE, R3.6:: geepack[50,51]). GEE models account for resampling bias in longitudinal measurements[52]; other regression models, like generalized linear models, overestimate the sample size and power by ignoring this bias. Unlike mixed effect models, which can account for resampling bias, GEE allows non-linear relations between the outcome and covariates, while accounting for correlation among repeated measurements from the same subject. Here we used GEE with an exchangeable correlation structure (assuming the within-subject correlation between two time-points is $\rho$). We log and z-score standardized each glycan and glyco-motif measurement to stabilize the variance and equalize the range. We also used the log of days postpartum (DPP) to linearize the relationship over time. The Wald test was used to measure the significance of secretor status contribution. For additional information and diagnostic statistics for specific regressions, see Supplementary Table 3a, b. All regression results can be found in Supplementary Fig. 10.

**Product-substrate ratio as a proxy for flux and estimating flux-phenotype associations**. To further isolate glyco-motif-specific effects from biosynthetic biases, we explored methods to control for the product-substrate relations. First, we extract the relative abundance of parent–child pairs of glyco-motifs in the substructure network; these are product-substrate relations like LNT and LSTb (Fig. 5e). Glyco-motif abundance represents the total substructure synthesized; therefore, when we examine the product-substrate ratio, we measure the total amount of the substrate substructure converted to the product substructure in the sample. Thus, the product-substrate ratio is a proxy for flux. Using logistic GEE regression modeling, similar to the approach used for testing substructure-phenotype associations, we can measure the influence of estimated flux between two glycans on secretor status; here we predicted secretor status from the estimated flux log(DPP). For additional information and diagnostic statistics, see Supplementary Table 3c.

**Glyco-motif abundance robustness and power analysis**. Similar to those used in Supplementary Fig. 9, GEE models were trained using either glyco-motif or whole glycan relative abundance. To stabilize the variance, equalize the range, and make the regressions comparable, we used a square root and z-score normalization on each glycan and glyco-motif measurement. Glyco-motif or relative glycan abundance was predicted from either DPP alone, secretor status alone, DPP + secretor status, or DPP + secretor status + DPP:secretor. To avoid biasing the analysis with misfit or uninformative models, models with small coefficients ($|coef| < 0.5$) or non-normal abundance distributions (Shapiro–Wilks $p < 0.001$) were removed. Model robustness measures including, coefficient magnitude ($n_{\text{glycan-stats}} = 39$, $n_{\text{motif-stats}} = 86$), standard error ($n_{\text{glycan-stats}} = 39$, $n_{\text{motif-stats}} = 86$), and marginal $R^2$ ($n_{\text{glycan-stats}} = 21$, $n_{\text{motif-stats}} = 47$) were used to compare model performance. Robustness measures from glycan-trained and glyco-motif-trained models were compared using a one-sided Wilcoxon rank-sum test with continuity correction. We validated these findings using a 10,000 iteration one-sided, two-sample bootstrapping $t$-tests (Rv3.6::nonpar::boot.t.test); bootstrapping $p$-values were less than or equal to Wilcoxon rank-sum $p$-values within 0.001. Finally, using the Rv3.6:: pwr::pwr.r.test v1.2.2 package, statistical power was predicted between $n = 5$ and $n = 200$ for the median and interquartile range of effect sizes observed in glyco-motif-trained and glycan-trained models.

**Substructure decomposition of published IgG N-glycosylation to distinguish known and unknown biosynthetic reactions**. We re-analyzed structural N-glycan data from IgG (Benadetti, 2017)[28]. IgG N-glycans were measured using liquid chromatography coupled with electrospray mass spectrometry (LC-ESI-MS). Pre-processing of these data was restricted to reformatting for input into Glycompare-compatible abundance matrix and structure annotation. Glycoprofiles were normalized to relative abundance. Substructure abundances and motif extraction were performed using an N-glycan thereby focusing analysis on biosynthetic motifs.

Using the IgG N-glycan data, we estimated partial correlation[53] between glycan abundances or between motif abundances. Previously, glycan abundance partial correlation was used to identify previously uncharacterized N-glycan biosynthetic reactions[28]. Here, we used motif abundance partial correlation and compared predicted power. Edges (partial correlations between glycans or motifs) were filtered for direct relations (structures differing by only one monosaccharide), split into known (True), and unknown (False) reactions. Partial correlation distributions were stratified by prior knowledge (True vs False), structure type used for partial correlation (glycan vs motif), IgG isoform (1, 2, or 4), and reaction type (B4GALT or ST6GALT1; manually annotated). A one-sided $t$-test was used to determine if motif abundance calculated partial correlations were higher than those calculated from glycan abundance in either previously known or unknown reactions.

**Substructure decomposition of published mucin-type O-glycans to clarify tumor-specific glycan epitopes**. We re-analyzed structural mucin-type O-glycan abundance[37]. Mucin-type O-glycans were originally measured by liquid

chromatography and mass spectrometry (LC-MS), structures were manually annotated using empirical masses from Unicarb-DB[24]. Pre-processing of these data was restricted to reformatting for input into Glycompare-compatible abundance matrix and structure annotation. Formatted data were normalized using probabilistic quotient normalization[54]. Substructure abundances and motif extraction were performed using a monosaccharide core for thereby focusing analysis on epitope motifs.

Using the mucin-type O-glycan data, we examined both the original glycan abundance data and the motif-level abundance decomposition. Glycan and motif structure abundance was compared across cancer and non-cancer samples using two-sample $t$-tests; $p$-values were multiple-test corrected using false discovery rate[55].

**Substructure decomposition of ganglioside glycolipids to compare abundance across tissues**. We re-anaylzed structural ganglioside glycolipid abundance[38]. Published abundance was pooled (summation) within ceramide types, from mouse eye, brain and blood. Glycosides abundance was originally measured by hydrophilic interaction liquid chromatography stratified mass spectrometry (HILC-MS), and HPLC with glycoside standards for structural identification. Pre-processing of these data was restricted to reformatting for input into GlyCompare-compatible abundance matrix and structure annotation. Formatted data were normalized using probabilistic quotient normalization[54]. Substructure abundances and motif extraction were performed using a lactose core thereby focusing analysis on biosynthetic motifs.

We examined abundance from two gangliosides (GD3 and GM2) and their corresponding lactose-based substructure abundance. Ceramide groupings include more than 42 or fewer than 35 Carbons ($C_{>42}$, $C_{<35}$), either 1 or 2 unsaturated bonds (1 unsat., 2 unsat.), or groups of specific ceramides with X:Y carbons and unsaturated bonds (e.g., 34:1, (36:1+38:1), or (40:1+40:2). Due to limited sample size, trends rather than formal statistics were used to compare abundance.

**Substructure decomposition of site-specific N-glycan compositions to enrich correlation structure**. We re-anaylzed compositional site-specific N-glycan abundance[22]. Intact site-specific N-glycan composition was measured using activated-ion electron transfer dissociation (AI-ETD), the log of localized spectra count for each site-specific composition was used to represent abundance. Pre-processing of these data was restricted to reformatting for input into a Gly-Compare-compatible abundance matrix and structure annotation. Formatted data were normalized using probabilistic quotient normalization[54]. Substructure abundances and motif extraction were performed using compositional monosaccharides thereby focusing analysis on epitope motifs.

Examining site-specific N-glycan compositional data from rat brain, we used a slightly modified method to compute compositional substructure abundance from compositional abundance. To calculate compositional substructure, we sum over larger and subsuming structures in a compositional network. Consider the compositional abundance of a structure: HexNac(p)Hex(q)Fuc(r). Instead of abundance of HexNAc = p, Hex = q, and Fuc = r, we examine the compositional abundance for all measurements where HexNAc$\geq$p, Hex$\geq$q, and Fuc$\geq$r. The network structure can be constrained to provide additional insight (e.g., Glyconnect Compozitor[56]), currently, the aggregation criteria remain simple. In analyzing these data, we explored trends in correlation between observed compositional vs compositional-substructure abundance.

**Reporting summary**. Further information on research design is available in the Nature Research Reporting Summary linked to this article.

## Data availability
The EPO N-glycan, IgG, glycolipid, mucin, and site-specific N-glycan abundance data reformatted and re-analyzed for this study as well as the HMO abundance data generated in this study have been deposited in Zenodo at https://doi.org/10.5281/zenodo.5072568. The data supporting this work is made available under a CC-BY 4.0 licence. Source data are provided with this paper.

## Code availability
We provide the Glycompare python library (v1.1.3) described in this work and example code used to perform analysis and generate figures are available through Github (https://github.com/LewisLabUCSD/GlyCompare/tree/v1.1.3) and Zenodo (https://doi.org/10.5281/zenodo.5072568). In addition to the Glycompare python library, we provide jupyter notebooks to generate our figures and analysis. Finally, a dockerized environment that supports Glycompare and all EPO and HMO analyses in the manuscript is available at https://doi.org/10.24433/CO.9148600.v1. The glycompare python package and examples are made available under an MIT licence.

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

## Acknowledgements

The authors thank Joshua Klein for guidance on using glypy and Dr. Philip Gordts for insightful comments. We also thank Elisa Benedetti for helping reproduce and extend her prior work. This work was conducted with support from the Novo Nordisk Foundation provided to the Technical University of Denmark (NNF10CC1016517, NNF20SA0066621: N.E.L.), NIGMS (R35 GM119850: N.E.L.), NICHD (R21 HD080682: L.B.), and USDA (USDA/ARS 6250-6001; M.W.H.). This work is a publication of the U.S. Department of Agriculture/Agricultural Research Service, Children's Nutrition Research Center, Department of Pediatrics, Baylor College of Medicine, Houston, Texas. The contents of this publication do not necessarily reflect the views or policies of the U.S. Department of Agriculture, nor does mention of trade names, commercial products, or organizations imply an endorsement from the U.S. government.

## Author contributions

B.B. and B.P.K. designed the work. B.B., B.P.K., A.W.T.C., J.T.S., Y.Z., A.K.Y., and N.E.L. performed data analysis. M.A.M., M.W.H., and L.B. provided HMO data. The manuscript was written by B.B., B.P.K., A.W.T.C., L.B., and N.E.L.

## Competing interests

The authors declare no competing interests.
