## [Peer Review File · Nature Communications]

REVIEWER COMMENTS

Reviewer #1 (Remarks to the Author):

This manuscript introduces a novel method for analyzing glycan profiles as usually obtained from glycomics mass spectrometry experiments. While this method is shown to be effective and statistically accurate, considering that the glycomics MS community is quite small, it is questionable whether this work would be of interest to the wider community.

This work is of high originality and the results are convincing; it would certainly influence the glycoinformatics community when approaching glycomics data. Since all the methods have been made publicly available, it can be assumed that this work can be reproduced by others.

Specific comments:

1. There should be some clarification that MALDI-TOF data basically produces compositional information, and that annotating the peaks with completely defined glycan structures makes several assumptions.

Thus care must be made when assigning enzymes to the biosynthetic pathways of these annotated glycans.

2. The term "glyco-motifs" may be misleading since the term Glycan motif is used often in glycan databases referring to glycan epitopes and glycan patterns with known names such as the lewis structures.

3. On page 11, line 236, the authors mention the increased conversion ratio of iGNT, which is derived from the substrate-product ratio. However, has any transcriptomics data or other experiments performed to confirm these results? A similar question applies to the statement on page 12, lines 251-252 regarding the maternal genetics affecting secretor status, as well as page 14, lines 301-302 discussing phenotype-specific reaction propensity providing insight into condition-specific synthesis

Minor comments:

- spelling of UniCarb-DB
- The term "a-fucosylated" may be misread; better to use "non-fucosylated"
- The English grammar made this manuscript difficult to read throughout. The grammar should be *thoroughly* checked by a native English speaker.

Reviewer #2 (Remarks to the Author):

The authors propose using glycan substructures, or intermediates, to address the issue of interdependence between measured different glycan species. This is a valuable approach that takes into account the various and often interdependent biosynthetic steps shared between different glycan structures in the complex hierarchical process of their biosynthesis. This leads to expansion of the original dataset which then has to be filtered for meaningful features. However, the steps towards creating a reduced dataset with a minimal subset of meaningful substructures are not completely clear, nor is it clear to which extent this will lead to a loss of information. The lines 164-169 mention the calculation of the weight of each monosaccharide, and removing monosaccharides with weight less than 51%, however it is not clear how this is done, how the threshold is chosen, and how this affects the data and low abundant features. The method section does unfortunately not contain an informative explanation. The quantitative aspect of this analysis is a black box, so I propose it should be explained in details and thoroughly reviewed by bioinformatics experts.

Although the authors mention other published approaches to address the sparsity (line 79-85) they show the benefits of their approach compared to the approach using individual N-glycans only. It is expected that sparsity of the dataset is reduced using this approach, however, the authors should demonstrate that their approach is superior to the other, published approaches using glycan motifs or epitopes, which also reduces sparsity and groups individual glycans by the similarities in their biosynthesis.

The comparison with glycan epitopes was performed on the O-glycan gastric cancer dataset, where the original publication does not demonstrate significant differences between tumors and control tissues. Although the authors of this manuscript claim they have discovered features that might distinguish the tissue types, this finding was not validated. They do not discuss the meaning of this finding, which glycosylation pathway it might be associated with and whether this makes sense in

terms of biology. This is disappointing considering they claim to be using a systems biology approach. The authors identify structures depleted in the tumors, core 2 with fucosylation and I branching, however, it is not clear how this was missed in the original publication. I would be interested to see if combining the derived traits from the original publication- core 2, I branching and core 2 Lewis fucosylation would give the same result. If not, that is worrying, and it poses a question whether the results from the tool are valid. If yes, it emphasizes the strongest feature of this tool, exploration of the interactions between epitopes/glycan motifs that might be missed in conventional analysis.

The authors claim that after creating the network of substructures the edges can be annotated by known biosynthetic pathways. However, it is important to account for the possibility of non-isomer-resolved glycomics data, that might mislead the pathway analysis and interpretation of the edges. Therefore, my suggestion is that detailed structural identification of all species with a clear isomeric separation is a prerequisite for the mentioned downstream analysis. In this way, a detailed network with all edges annotated by glycosyltransferase specificities will be possible, and insightful for the interpretation of the perturbed glycosylation profiles. It is not clear how ambiguous or unknown linkage information can affect the downstream analysis. Importantly, it is disappointing that the edges of the network were not annotated with relevant biosynthetic knowledge although authors claim to use a systems biology approach. The authors should present the biosynthetic rules they have used for creating the network edges in details.

Building the substructure network is the key point of the tool but description of the process is very cryptic. If a logic of the vectorization and generation of a substructure vector is clear, the only information provided on the substructure network generation is that it is a directed acyclic graphs (DAG) where “each node denotes a glycan substructure” and “ the edges in the substructure network were annotated with known synthetic rules”. Even a library used for generation of the DAG is not mentioned. DAG is a graphical model of the data and as a model it requires tuning/optimization, yet not a single word is spent on this. How do we suppose to know that at every presented application the optimal data driven model was chosen and not the one which according to the authors proves the point better? The output strongly depends on the substructure network, whilst analysis in the “classical” way, the one they criticize, may be considered more robust as it mainly depends on the raw data and not on extensive tuning.

In the example with EPO the authors claim that standard method of the glycan profile analysis fails to get a “right” clustering of the glycoengineered CHO cell lines. In doing so they appeal to a visual assessment of the two cluster solutions comparing a clustering of the glycoprofiles based on the glycans and a one on the glyco-motifs. The authors do not bother themselves with a description of the clustering algorithm, they call it a “standard algorithm”, but a good guess would be that the hierarchical clustering (HC) was used. While HC can work on the sparse data, it is clearly sub-optimal for such applications (the authors admit it in the text), but then a difference between the solutions is not due to the superior performance of the advertised tool, but due to a suboptimal or

inappropriate clustering method used on the sparse data. If the authors had chosen for applying a clustering algorithm which can deal with the sparsity, a “right” solution might have been obtained on glycan based glycoprofiles. Moreover, a comparison of the cluster solutions should not be based on the visual assessment only; one could use one of the dissimilarity measures and compare it between the solutions or compare the general agreement between the clustering solutions using e.g. the adjusted Rand Index.

The main terms on which the authors build their argumentation and the "raison d'être" of their tool are poorly if at all defined. How is the term “interdependence” different from the co-linearity? Is the a numerical measure of the interdependence?

Reviewer #3 (Remarks to the Author):

With the increasing output of high quality glycan profiling data by the biomedical community, there is a clear need for a bioinformatic formalism for comparison of such data among sets of biological variants. This is because glycosylation created in the secretory pathway consists of a set or related structures created by biosynthetic reactions that do not go to completion. The authors address this challenge by accounting for shared biosynthetic steps for measured glycans. This approach addresses effectively the need to correct for sparsity and non-independence in glycan distributions. They demonstrated their approach appropriately using engineered erythropoietin samples as a well-understood benchmark. This allowed the authors to identify knockout-specific effects on the glycan profiles. This is an important example that illustrates the potential value for studies of disease models that test the effects of specific gene manipulation. They then studied more complex examples of human milk glycans, gangliosides, and tissue-specific glycan profiles. The substructure-oriented approach appears to be quite useful for supporting conclusions regarding the genetic significance of glycan profiles in biological systems. The manuscript has been vetted thoroughly in previous reviews and appears appropriate for publication in Nature Communications.

Reviewer #1 (Remarks to the Author):

This manuscript introduces a novel method for analyzing glycan profiles as usually obtained from glycomics mass spectrometry experiments. While this method is shown to be effective and statistically accurate, considering that the glycomics MS community is quite small, it is questionable whether this work would be of interest to the wider community.

This work is of high originality and the results are convincing; it would certainly influence the glycoinformatics community when approaching glycomics data. Since all the methods have been made publicly available, it can be assumed that this work can be reproduced by others.

We appreciate your candor and the compliments on the quality of the work. We anticipate that by addressing major challenges in the handling of glycomics data, that this work, with other novel methods being developed for big data analytics in glycomics, that there will be an increased surge in the application of glycomics data analysis in the broader scientific literature.

Specific comments:

1. There should be some clarification that MALDI-TOF data basically produces compositional information, and that annotating the peaks with completely defined glycan structures makes several assumptions.

Thus care must be made when assigning enzymes to the biosynthetic pathways of these annotated glycans.

We agree with this concern and have added additional clarification to avoid suggesting we know the “true” identity of the glycan. GlyCompare is a method for downstream analytics, so it’s important that users are aware of the assumptions. For our tool, we hope to improve what is done with annotated glycoprofiles after they are annotated by experts and the constantly improving software suites being developed by the MS community.

To clarify this to the readers, we have added text in the discussion to clarify the purpose and remaining uncertainties: “Discussion section: 387-390: “Like any analytical pipeline, GlyCompare is sensitive to upstream analysis (e.g., mass spectrometry methods measure the mass-to-charge ratios of glycans and their fragments, and thus require expert annotation to assign structures). Therefore, GlyCompare will continue to improve with advances in glycoprofile structure annotation quality.”

2. The term "glyco-motifs" may be misleading since the term Glycan motif is used often in glycan databases referring to glycan epitopes and glycan patterns with known names such as the lewis structures.

We were also concerned with this potential confusion. In the manuscript, we distinguish between glyco-motifs (frequently occurring substructures whose frequency indicates biosynthetic or function purpose) and glycan epitopes (a type of glyco-motifs appearing at the glycan branch-terminus that typically has functional implications through lectin recognition).

To improve clarity and avoid confusion, we have defined these in the main text and added text centrally in the Figure 2 caption to clarify the distinction. (lines 798-807) “Throughout the manuscript, glycan is referred to complete and secreted monosaccharide polymer; a glycan substructure is referred to a complete or incomplete monosaccharide polymer observable within at least one secreted glycan; a glycan motif (glyco-motif) is referred to an enriched functional glycan substructure for a dataset or biological process. Note that both glycan epitopes (typically terminal glycan substructures recognized by lectins) and glycan cores (biosynthetic glycan substructures common to select types (e.g. N- or O-glycosylation) or modes (e.g. complex or high-mannose) of biosynthesis) are glyco-motifs as they are biologically functional, interpretable and will be enriched in datasets selecting for specific glycan presentation of biosynthesis. Glycompare core methods are explained at length in the Methods section”

3. On page 11, line 236, the authors mention the increased conversion ratio of iGNT, which is derived from the substrate-product ratio. However, has any transcriptomics data or other experiments performed to confirm these results? A similar question applies to the statement on page 12, lines 251-252 regarding the maternal genetics affecting secretor status, as well as page 14, lines 301-302 discussing phenotype-specific reaction propensity providing insight into the condition-specific synthesis

Though the primary focus of the analysis was to demonstrate a potential use case for GlyCompare, we do not want to overstate or misrepresent results. We thank the reviewer for reminding us of the importance of validation.

Regarding the iGNT claims, indeed, we have access to unpublished RNA-Seq data that shows that the enzymes in charge of the elongation are upregulated, contributing to the increased conversion ratio of iGNT. In CHO cell lines, the B3gnt family (B3gnt2, B3gnt5, B3gnt9) is primarily responsible for poly-LacNac elongation (also, the B3gal family (B3galt1, 4, 6) contribute to other types of elongation). We found that B3gnt2 and B3gnt9 are most relevant here since B3gnt5 was not detected in our samples). Below we show the results from the relevant knockout clones for which we have access to data.

- For St3gal knockouts, both single and double knockout confirm the upregulation of the enzymes that contribute to the elongation. St3gal4 single knockout can lead to upregulation of the B3gnts (B3gnt2,9) and B3galt1,4,6. Similarly, St3gal6 single knockout leads to upregulation of the B3gnts (B3gnt2,9) and B3galt1,6). St3gal4/6 double knockouts also lead to upregulation of B3gnt2,9 and B3galt1. The B3gnts were consistently modestly upregulated in the St3gals knockout samples.

(Note: the star indicates the significance (FDR adjusted p-values) of DE testing: '*': $p < 0.05$, '**': $p < 0.01$, '***': $p < 0.001$)

We show the data here, but the data is part of a larger collaborative study wherein RNA-Seq was generated from cell lines provided by Prof Henrik Clausen (the cell lines from which the glycoprofiles studied here were obtained and previously published). The manuscript for the larger transcriptomic study is under preparation, so we are not able to publish the data at this moment. However, the data will be made available once the other collaborative publication is accepted.

For the similar question regarding the HMO analysis, the exact isozymes catalyzing these reactions have not been experimentally determined at this moment. But fortunately, we were able to uncover plausible confirmation in a recent prominent publication (10.1093/jn/nxy175). We have added text to the corresponding HMO sections noting the consistency between our findings and the prior publication. "A secretor-elevated conversion rate from LSTb to DSLNT is consistent with observing elevated X62 and secreted LSTb in non-secretor milk (Fig. 5a-b)³⁶,"
 Line 302-303.

Minor comments:

- spelling of UniCarb-DB

Resolved

- The term "a-fucosylated" may be misread; better to use "non-fucosylated"

Resolved

- The English grammar made this manuscript difficult to read throughout. The grammar should be *thoroughly* checked by a native English speaker.

Reviewer #2 (Remarks to the Author):

The authors propose using glycan substructures, or intermediates, to address the issue of interdependence between measured different glycan species. This is a valuable approach that takes into account the various and often interdependent biosynthetic steps shared between different glycan structures in the complex hierarchical process of their biosynthesis. This leads to expansion of the original dataset which then has to be filtered for meaningful features.

We appreciate the kind words in regard to the value of this work.

1. However, the steps towards creating a reduced dataset with a minimal subset of meaningful substructures are not completely clear, nor is it clear to which extent this will lead to a loss of information. The lines 164-169 mention the calculation of the weight of each monosaccharide, and removing monosaccharides with weight less than 51%, however it is not clear *how this is done, how the threshold is chosen*, and how this *affects the data and low abundant features*. The method section does unfortunately not contain an informative explanation. The quantitative aspect of this analysis is a black box, so I propose it should be explained in details and thoroughly reviewed by bioinformatics experts.

We thank the reviewer for their concern and for pointing out the disconnect between our methods and their explanation. We hadn't adequately explained the method in the main text. In particular the separate methods for glyco-motif ("minimal subset of meaningful substructures") and representative structure ("monosaccharides with abundance above 51% in a glycan or substructure clusters") identification were less clearly described in the main text. We have revised the text accordingly, with an in depth description of the approach in the methods (section 2 and 3). We also present it visually, as illustrated in Figure 2. We have also added text to the caption to help mitigate this disconnect for future readers.

Specifically, the reviewer says, "...the steps towards creating a reduced dataset with a minimal subset of meaningful substructures ...". In the paper, we explain "...a parent-child substructure pair that has the same abundance (solid arrow), can be merged" (lines 532). If a parent-child has the same abundance, we can conclude that they carry the same information.

We have also added text to the discussion to indicate when users shouldn't expect information loss "when biosynthesis is not hierarchical and acyclic." (line 394)

Regarding the "...calculation of the weight of each monosaccharide, and removing monosaccharides with weight less than 51%..." (line 164). This happens after the clustering (line 163), so it doesn't directly relate to "a reduced dataset with a minimal subset of meaningful substructures" that happens before the clustering (line 160). Our concept of "representative substructures" is a heuristic approach used to summarize the common structure information from a substructure cluster. Using 51% as the threshold is like getting the value that is slightly above the mean of a group of digits. We have added text to the methods "a heuristic and flexible parameter" to clarify the nature of this cutoff (line 551).

2. Although the authors mention other published approaches to address the sparsity (line 79-85) they show the benefits of their approach compared to the approach using individual N-glycans only. It is expected that sparsity of the dataset is reduced using this approach, however, the authors should demonstrate that their approach is superior to the other, published approaches using glycan motifs or epitopes, which also reduces sparsity and groups individual glycans by the similarities in their biosynthesis.

The reviewer makes an important point here. We agree that this must be demonstrated beyond single N-glycan structures. To that end, we show the de-sparsing function in O-type HMOs (Supplementary Fig 9) and site-specific N-glycan compositional data (Supplementary Fig 13). The Benedetti paper (Nat Comm, 2017), like ours, contains post-annotation substructure analysis. Towards addressing the reviewer's concern, we compare our ability to recover biosynthesis reactions by comparison to the Benedetti 2017 paper (see Results section on "increased statistical power" line 330-348 and Supplementary Fig 11) "To further probe the increased statistical power, we compared our approach to another statistically-driven network approach. Benedetti et al. 2017 demonstrated that novel glycan biosynthetic reactions could be resolved using partial correlation²⁸. Using the Benedetti data, we computed partial correlation for glycan abundance and with GlyCompare-computed linkage-specified substructure abundance. We compared the partial correlation between glycans or substructures across "true," known reactions and "false," uncharacterized reactions (as specified in the Benedetti supplement). Partial correlations across known reactions between GlyCompare-computed substructures were significantly higher than partial correlations between corresponding glycan abundances (Supplementary Fig. 11). Partial correlation across known reactions was elevated for substructure abundance in all IgG isoforms (One-sided t-test, $p < 0.0039$), as well as responses performed by B4GALT1 and ST6GAL1 (One-sided t-test, $p < 1.1 \times 10^{-4}$). Interestingly, the lowest partial correlations across "true" reactions between substructures were substantially higher than corresponding glycan correlations. The higher floor for substructure correlations suggests that substructure abundances may increase positive predictive value (Supplementary Fig. 11). Finally, while correlation increased between "true" associated substructures, correlations across uncharacterized reactions were close to zero and indistinct from glycan correlations across the same reactions. Thus, using GlyCompare for glyco-motif-level analysis can substantially increase the robustness and statistical power in glycomics data analysis since it allows for comparing different glycans who share biosynthetic steps."

Supplementary Figure 11 | Partial correlation between known and unknown biosynthesis

reactions in N-glycosylation. In panel **a** partial correlations are split by IgG isoform while panel **b** shows partial correlations split by related glycosyltransferases. Glycan abundance data (MALDI-TOF) from Benedetti et. al. 2017⁴ was realized to compute partial correlation between glycan abundance (as in the original paper, blue) and glycompare-computed linkage-specified substructure abundance (red). Partial correlations were stratified by prior knowledge, those known and previously characterized were designated the “true” (T) reactions (Benedetti et. al. 2017 Supplemental Dataset S1), while the other uncharacterized reactions were designated “false” (F).

Some issues made a direct comparison to other methods difficult. The Ashwood and Rademacher methods were substructure-oriented but developed for raw MS data analysis while our approach operates on processed data. These papers are more focused on annotation and do not make claims on sparsity; therefore, a direct comparison may be possible but potentially misleading given the methods were designed for different applications. The Klein and Hosoda methods focused on specific software functions and did not make claims on sparsity, interdependence, or biosynthesis. The Alocci publication is focused on visualization, not statistical significance. The Sharapov analysis was a similar post-annotation analysis, but because of the similarity in data sources and preparation to the Benedetti analysis, these results would not have provided additional information. Finally, the Klammer approach is specific to lectin arrays. We suspect GlyCompare is extensible to motif-discovery from lectin arrays but we have not completed this investigation, and we look forward to thoroughly pursuing this in a follow-up study.

3. The comparison with glycan epitopes was performed on the O-glycan gastric cancer dataset, where the original publication does not demonstrate significant differences between tumors and control tissues. Although the authors of this manuscript claim they

have discovered features that might distinguish the tissue types, this finding was not validated. They do not **discuss the meaning of this finding, which glycosylation pathway it might be associated with and whether this makes sense in terms of biology**. This is disappointing considering they claim to be using a systems biology approach. The authors identify structures depleted in the tumors, core 2 with fucosylation and I branching, however, it is not clear how this was missed in the original publication. I would be interested to see if **combining the derived traits from the original publication- core 2, I branching and core 2 Lewis fucosylation would give the same result**. If not, that is worrying, and it poses a question whether the results from the tool are valid. If yes, it emphasizes the strongest feature of this tool, exploration of the interactions between epitopes/glycan motifs that might be missed in conventional analysis.

Thank you for these insightful suggestions. We have added some commentary on the biosynthetic implications by comparing to prognosis-stratified survival in TCGA using the TIDE database. It seems that B3GNT3, necessary to synthesize the initial structure of those we note as depleted, is depleted in gastric cancers with a worse prognosis. We also compared our results and found them consistent with a later publication in gastric cancer glycosylation. **Line 365-367, "Also consistent with the decrease in bi-GlcNAc core-2 structures in gastric cancer, low expression of B3GNT3 in stomach cancer is significantly associated with decreased overall survival³⁸. B3GNT3 is necessary for adding the second GlcNAc to core 2 structure³⁹ and therefore upstream of all significantly depleted structures (Fig. 7); B3GNT3 depletion could explain the observed differential glycosylation."**

We agree that it is counter-intuitive that we found this trend while the original publication did not. We argue that our increased accuracy is a feature of our tool. We effectively look at previously unobserved values predicted from secreted abundance: the unobserved abundances within the Golgi. Considering we are looking at previously unexamined metrics and are not looking at the whole-glycan level (allowing us to see enrichments in common substructures), it is less surprising that we see novel trends.

4. The authors claim that after creating the network of substructures the edges can be annotated by known biosynthetic pathways. However, it is important to *account for the possibility of non-isomer-resolved glycomics data*, that might mislead the pathway analysis and interpretation of the edges. Therefore, my suggestion is that detailed structural identification of all species with a clear isomeric separation is a prerequisite for the mentioned downstream analysis. In this way, a detailed network with all edges annotated by glycosyltransferase specificities will be possible, and insightful for the interpretation of the perturbed glycosylation profiles. It is not clear how ambiguous or unknown linkage information can affect the downstream analysis. Importantly, it is disappointing that the edges of the network were not annotated with relevant biosynthetic knowledge although authors claim to use a systems biology approach. The authors should present the biosynthetic rules they have used for creating the network edges in details.

We agree that the upstream processing will be crucial to the interpretability of the substructures. We are trying to avoid strict limitations on using the approach as different functions are

appropriate for different data types. For example, as you mention, usage of all functions is inappropriate without isomer-resolved data. But, as demonstrated in Supplementary Fig 13, using only the substructure decomposition can be helpful for compositional data. We have included some clarification in the discussion on what data-types are appropriate for different GlyCompare functions and when information loss can occur: Line 387-396, “Like any analytical pipeline, GlyCompare is sensitive to upstream analysis (e.g., mass spectrometry methods measure the mass-to-charge ratios of glycans and their fragments, and thus require expert annotation to assign structures). Therefore, GlyCompare will continue to improve with advances in glycoprofile structure annotation quality. Going forward, we hope to include multiple methods for aggregating abundance over substructures, including aggregation using multiple functions (besides addition) over fully or partially specified biosynthetic networks. While summing abundance for all subsumed substructures works well, manual reaction specification can help avoid information loss when biosynthesis is not hierarchical and acyclic or glycans are not increasing in size. When these limitations are acknowledged, the current version of glycompare has demonstrated some exciting capabilities.”

We have also included a supplemental data file containing the complete substructure network and the biosynthesis reactions referenced in the text (Supplementary Data S1).

5. Building the substructure network is the key point of the tool but description of the process is very cryptic. If a logic of the vectorization and generation of a substructure vector is clear, the only information provided on the substructure network generation is that it is a directed acyclic graphs (DAG) where “each node denotes a glycan substructure” and “the edges in the substructure network were annotated with known synthetic rules”. Even a library used for generation of the DAG is not mentioned. DAG is a graphical model of the data and as a model it requires tuning/optimization, yet not a single word is spent on this. How do we suppose to know that at every presented application the optimal data driven model was chosen and not the one which according to the authors proves the point better? The output strongly depends on the substructure network, whilst analysis in the “classical” way, the one they criticize, may be considered more robust as it mainly depends on the raw data and not on extensive tuning.

We clarify that the “directed acyclic graph” (lines 517-520) is a data structure and mathematical concept, as opposed to a model. “Thus, it’s unclear what is meant by the comment that a graph model requires training; a graph is simply a mathematical construct used here to communicate the network with nodes as “substructures” and edges “known reactions”. Thus, we have fully described the DAG.

We have clarified in the methods that the software package (networkx) was used to manage these structures. The network generation code is all written by hand, and now provide a detailed explanation right after mentioning the “directed acyclic graph” in the method. “Given the substructure set S, the root node starts from the monosaccharides or a defined root core structure, and a child node is a substructure that has only one monosaccharide added to its parent node.” A tool example is visualized in Figure 2.

Finally, we acknowledge the concern about robustness given the flexibility of the substructure network. In particular, we were also concerned about this during the design of this project. For that reason, we separated the calculation of substructure abundance (keeping this predictable and straightforward) and the glyco-motif selection (which uses the substructure network to select the most interesting substructures). There is no augmentation to the core data during motif selection that could decrease the consistency of these results. Substructure abundance should be as reproducible as its upstream data; motif selection simply allows the user to focus on the relevant substructures necessary to understand their dataset. Due to the importance of this point, we have added text to the discussion saying “In another word, substructure abundance is automatically transformed from the upstream data; motif selection simply allows the user to focus on the least substructures necessary to understand their dataset.” (lines 381-383)

6. In the example with EPO the authors claim that standard method of the glycan profile analysis fails to get a “right” clustering of the glycoengineered CHO cell lines. In doing so they appeal to a visual assessment of the two cluster solutions comparing a clustering of the glycoprofiles based on the glycans and a one on the glyco-motifs. The authors do not bother themselves with a description of the clustering algorithm, they call it a “standard algorithm”, but a good guess would be that the hierarchical clustering (HC) was used. While HC can work on the sparse data, it is clearly sub-optimal for such applications (the authors admit it in the text), but then a difference between the solutions is not due to the superior performance of the advertised tool, but due to a suboptimal or inappropriate clustering method used on the sparse data. If the authors had chosen for applying a clustering algorithm which can deal with the sparsity, a “right” solution might have been obtained on glycan based glycoprofiles. Moreover, a comparison of the cluster solutions should not be based on the visual assessment only; one could use one of the dissimilarity measures and compare it between the solutions or compare the general agreement between the clustering solutions using e.g. the adjusted Rank Index.

We definitely agree that the term “standard” was ambiguous and should have been clarified. Thus, we have added an explicit description of our clustering method in the caption of Figure 3. We further agree that if we had used hierarchical clustering, it indeed would have been suboptimal and could have explained our results. We instead had used a correlation-based distance metric in our clustering, which (when sparsity is not a symptom of information compression) can be more robust to sparsity. Now that we have clarified our methods, we hope the reviewer will agree this concern has been addressed.

The reviewer raised an important concern regarding our “visual assessment.” To address this concern, we include the similarity matrix (correlation table) in Supplementary Figure 1. Additionally, as the reviewer suggested, we have included a bootstrapping-based robustness test Supplementary Figure 4-5.

Supplementary Figure

Supplementary Figure 1 | The glycoprofile clustering table with the original glycans.

This is the clustering of sixteen glycoprofiles based on glycans. Since most of the glycans only exist in a few glycoprofiles, so the clustering mainly focuses on the present/absent of the glycans (clusters 3-10), which means the information of structural similarity tend to be ignored in the clustering. This would drastically limit their analytic power due to the sparsity of comparable consensus glycans. The correlation distance tables are also provided for both glycan profiles and glyco-motif profiles.

Supplementary Figure 4 | Robustness of glyco-motifs clusters

This is the cluster of glyco-motif vector for EPO data. The robustness gives the criteria of how many substructure clusters should be generated. The clusters are distinguished if AU(red)=100 (approximately unbiased probability value $p < 0.01$) and then BP(green) (Bootstrap Probability) > 15 . The big block is further breakdown. We get 35 clusters in our EPO data. See Maechler, M., Hornik, K.(2019). cluster: Cluster Analysis Basics and Extensions. R package version 2.1.0.

Supplementary Figure 5 | The clustering robustness.

The robustness is measured with BP (Bootstrap Probability, Figure S4). BP is a measure of the cluster robustness suggesting a significant similarity within clusters and thereby mitigating some challenges of clustering reproducibility. Glyco-motifs abundances showed higher BP than clustering with whole-glycan abundance profiles. In the whole-glycan profile clusters, wild-type (WT) glycoprofiles are closer to the double-knockouts with highly-perturbed glycoprofiles. Double-knockouts with predominantly determined to be strongly perturbed and therefore should not cluster with wild-types in a biologically meaningful clustering. As such, we believe the glycan clustering (which clusters WT with double knockouts) is less interpretable than the glyco-motif clustering which does not include the WT/double-knockout grouping.

The similarity matrix showed the clear dissimilarity across groups, and the bootstrapping results corroborated the original clustering results. Because we used a sparsity-robust distance metric, confirmed sparsity was maintained in the correlation matrix, and confirmed our result with bootstrapping; we are highly confident that our clusters in Fig 1 and Fig 3 honestly describe the challenges of working with glycoprofile data. While we are sympathetic to the reviewer's

concerns, we are also aware that clustering is fundamentally heuristic. The leaders in the field do not have a rigorous statistical answer to choosing and validating clusters (<https://cran.r-project.org/web/packages/mclust/vignettes/mclust.html>). For this reason, we try not to focus too much on the “true” clusters but rather note the change in “natural segregation.” While the clustering is an interesting visual, it is more of an illustration than a result. The clusters are there to help explain the intuition of products to follow.

7. The main terms on which the authors build their argumentation and the "raison d'être" of their tool are poorly, if at all defined. How is the term “interdependence” different from the co-linearity? Is the a numerical measure of the interdependence?

In preparing this manuscript, we discussed at length how to describe the relationship between glycans--as substrates, products, and secretions. Ultimately, we chose two deliberately general terms (interdependent and non-independent). Because glycans are substrates for each other, subject to nested and propagating non-linear constraints, non-independence is axiomatic to the system. Interdependent is a general and imprecise term that includes many well-defined terms describing correlation or association.

However, the reviewer makes an important point that interdependence can be described by co-linearity since the co-linearity explicates some variance in one glycan being explained by another. But, because the manuscript is not a thorough characterization of the oddities of glycan abundance distributions, we were wary of asserting as to the nature of the disturbance. Additionally, we do not claim to change non-independence, interdependence, or co-linearity. Instead, we provide substructure abundance as a lens for examining the biosynthetic structures that lead to the emergence of non-independence in whole-glycan and substructure abundance

Reviewer #3 (Remarks to the Author):

With the increasing output of high quality glycan profiling data by the biomedical community, there is a clear need for a bioinformatic formalism for comparison of such data among sets of biological variants. This is because glycosylation created in the secretory pathway consists of a set or related structures created by biosynthetic reactions that do not go to completion. The authors address this challenge by accounting for shared biosynthetic steps for measured glycans. This approach addresses effectively the need to correct for sparsity and non-independence in glycan distributions. They demonstrated their approach appropriately using engineered erythropoietin samples as a well-understood benchmark. This allowed the authors to identify knockout-specific effects on the glycan profiles. This is an important example that illustrates the potential value for studies of disease models that test the effects of specific gene manipulation. They then studied more complex examples of human milk glycans, gangliosides, and tissue-specific glycan profiles. The substructure-oriented approach appears to be quite useful for supporting conclusions regarding the genetic significance of glycan profiles in biological systems. The manuscript has been

vetted thoroughly in previous reviews and appears appropriate for publication in Nature Communications.

We appreciate the positive comments from this reviewer.

REVIEWERS' COMMENTS

Reviewer #2 (Remarks to the Author):

The authors prepared a very thorough revision which considerably increased the clarity and expected impact of the manuscript.